# IMPROVING RANDOM-SAMPLING NEURAL ARCHITECTURE SEARCH BY EVOLVING THE PROXY SEARCH SPACE

## ABSTRACT

Random-sampling Neural Architecture Search (RandomNAS) has recently become a prevailing NAS approach because of its search efficiency and simplicity. There are two main steps in RandomNAS: the training step that randomly samples the weight-sharing architectures from a supernet and iteratively updates their weights, and the search step that ranks architectures by their respective validation performance. Key to both steps is the assumption of a high correlation between estimated performance(i.e., accuracy) for weight-sharing architectures and their respective achievable accuracy (i.e., ground truth) when trained from scratch. We examine such a phenomenon via NASBench-201, whose ground truth is known for its entire NAS search space. We observe that existing RandomNAS can rank a set of architectures uniformly sampled from the entire global search space(GS), that correlates well with its ground-truth ranking. However, if we only focus on the top-performing architectures (such as top 20% according to the ground truth) in the GS, such a correlation drops dramatically. This raises the question of whether we can find an effective proxy search space (PS) that is only a small subset of GS to dramatically improve RandomNAS's search efficiency while at the same time keeping a good correlation for the top-performing architectures. This paper proposes a new RandomNAS-based approach called EPS (Evolving the Proxy Search Space) to address this problem. We show that, when applied to NASBench-201, EPS can achieve near-optimal NAS performance and surpass all existing state-of-the-art. When applied to different-variants of DARTS-like search spaces for tasks such as image classification and natural language processing, EPS is able to robustly achieve superior performance with shorter or similar search time compared to some leading NAS works. Our code is available at https://github.com/IcLr2020SuBmIsSiOn/EPS

## 1 INTRODUCTION

Neural architecture search (NAS) has been successfully utilized to discover novel DNN architectures in complex search spaces and outperformed human-crafted designs. Early NAS works like NASNet (Zoph et al. (2018)) and AmoebaNet (Real et al. (2019)) used reinforcement learning or evolutionary algorithms to search for the DNN architectures by training a substantial amount of independent network architectures from scratch. Although these searched architectures can deliver high accuracy, they come with tremendous computation and time costs. Therefore, researchers gradually shift their focuses to one-shot NAS, which is more efficient and can deliver satisfying outputs within a few GPU-days. There are two main types of one-shot NAS. One is the differentiable NAS (DNAS), such as Liu et al. (2019b); Cai et al. (2018); Xie et al. (2018); Dong & Yang (2019b); Xu et al. (2019); Chen et al. (2019a), which uses a continuous relaxation of the architecture representations and introduces architecture parameters to distinguish the architectures. The other is Random-Sampling NAS (RandomNAS), such as Li & Talwalkar (2019); Chen et al. (2019b); Zhang et al. (2020); Guo et al. (2019); Bender (2019); Yang et al. (2020). RandomNAS approaches typically have two phases: (1) Training phase: in each iteration, RandomNAS randomly samples one architecture or a set of architectures and updates their shared weights in the supernet; (2) Search phase: after supernet training, the desired architectures are selected based on their performance ranking on the validation dataset

using inherited weights from the supernet, which is called weight-sharing performance. Finally, the selected architectures will be retrained from scratch to get their actual (retrained) performance for deployment. Compared to DNAS, RandomNAS usually consumes less GPU memories by partially updating the weights. Also, it generates multiple target architectures, while DNAS generally retrieves a single architecture based on the maxima of the representation distribution. There are, however, two major drawbacks preventing RandomNAS from achieving higher search efficiency. First, although RandomNAS achieves a promising ranking correlation between the weight-sharing estimation and the retrained performance over all architecture candidates, it delivers a low ranking correlation among "good" architectures (*e.g.*, top-20%-performing architectures among the search space) that researchers are more interested in. Second, by following the RandomNAS approach, smaller network architectures (with less parameters) tend to converge faster than the larger ones, which significantly degrades the ranking correlation.

To address the drawbacks of RandomNAS, we first introduce a proxy search space(PS): a subset of the architectures flexibly sampled from the global search space(GS) to study the features of RandomNAS. We then evaluate the RandomNAS with the proposed PS using the NASBench-201 benchmark (Dong & Yang (2020)) and notice two interesting phenomena: (1) When uniformly sampling a PS from a global search space, RandomNAS in the PS will maintain a similar ranking correlation with the one in the GS, even the size of the PS is extremely small (*e.g.* 16 architectures). (2) If the PS consists of the "good" architectures, the PS-based search can significantly improve the ranking correlation among the PS compared to the RandomNAS trained in GS and validated in PS.

Based on these observations, the PS constructed from "good" architectures can help overcome the first drawback of RandomNAS to search for more promising architectures. So, the remaining question is how to find a suitable PS containing sufficient "better" architectures? In this paper, we consider it a natural selection problem and solve it by the evolutionary algorithm. The architectures in the initial PS are iteratively evolved and gradually upgraded, while the average ranking of the architectures in the PS is improved. Meanwhile, it also helps improve the ranking correlation of the PS. We propose a new RandomNAS approach, named Evolving the Proxy Search Space (EPS). EPS runs in three stages iteratively: **Training** the supernet by randomly sampling from a PS; **Validating** the architectures among the PS on a subset of the validation dataset in the training interval; **Evolving** the PS by a tournament selection evolutionary algorithm with the aging mechanism. In this way, EPS gradually includes more high quality architectures in the proxy search space, while improves its ranking correlation. To solve the second issue in which smaller architectures converge faster than larger ones, we introduce a simple model-size-based regularization in the final selection stage. Our result on NASBench-201 shows 17.2% improvement in ranking correlation measured by Spearman's $\rho$ by adding the regularization on NASBench-201.

In the experiments, we demonstrate that EPS delivers a near-optimal performance on NASBench-201. We also extend the EPS on DARTS search space. By using the 5-search-runs measurement, EPS demonstrates a robust search ability compared with the previous works in 8 hours search time with little hyper-parameters fine-tuning effort. Also, EPS is evaluated in 4 DARTS sub search space (Zela et al. (2019)) using 3 datasets, on which DARTS easily fails. EPS surpasses the DARTS-ADA and DARTS-ES on most cases and can often find the global state-of-the-art architectures. EPS also shows a high performance on a language modeling task which consolidates the generalization ability and robustness of EPS.

## 2 RANDOMNAS ON NASBENCH-201

In this section, we present two major drawbacks we found in the existing RandomNAS methods and investigate the ranking correlation using a proxy search space. Proxy search space (PS) is a subset of the global search space (GS) and the RandomNAS is more flexible to be analyzed by training in the PS including different architectures from the GS. We run the following experiments on NASBench-201, which is a unified and fair benchmark designed for NAS algorithm evaluation and contains the ground-truth architecture accuracy on three datasets. The GS of NASBench-201 contains 15,625 architectures and they are constructed by 4 nodes and 5 associated operation options. (Please refer to Fig. 1 in Dong & Yang (2020) for details.)

## 2.1 RANKING CORRELATION IN DIFFERENT PROXY SEARCH SPACES

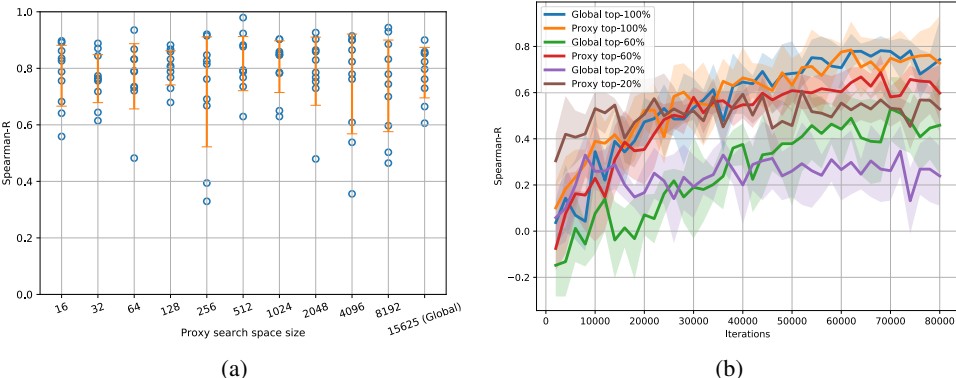

(a)                                                                 (b)

Figure 1: (a) The comparison of the Spearman's $\rho$ across 10 PS and the GS. The architectures in the PS are uniformly sampled from the GS. (b) Six different scenarios of Spearman's $\rho$ through the RandomNAS training. "Global top-x%" stands for the RandomNAS trained in the GS and evaluated by architectures in the the top-x% of the GS. "Proxy top-x%" stands for the RandomNAS trained and evaluated in a PS which consists of architectures in the top-x% of the GS.

First, we want to study whether RandomNAS in a PS with architectures uniformly sampled from the GS can achieve a higher ranking performance than the RandomNAS in the GS. We design an experiment to train the RandomNAS in the PS with 10 different sizes from $2^4$ to $2^{13}$ independently and compare the results with the RandomNAS in the GS. The experiment for each scenario is conducted 10 times with different random seeds ($10\times11$ total runs). We calculate the Spearman's $\rho$ ranking correlation between the RandomNAS validation loss and the ground-truth test accuracy with 16 architectures randomly sampled from each search space. Fig. 1a shows the raw data with $\pm$ one standard deviation error bar for different proxy search space. The ranking correlations are close and there is no significant difference among them through statistical testing with a Pearson coefficient as -0.10. We observe that when the architectures are randomly sampled from the global search space, RandomNAS in a PS maintains a similar global-ranking performance with RandomNAS in the GS.

Second, although RandomNAS trained in the GS achieves 0.783 Spearman's $\rho$ when validating in the GS, it shows a poor Spearman's $\rho$ (0.347) in top-20% of the GS (which consists of top-20%-performing architectures ranked by ground-truth accuracy from the GS). Hence, we'd like to see if RandomNAS in a PS can have a better ranking performance. Fig. 1b shows another six different scenarios, demonstrating the ranking correlation differences when sampling from GS or PS. The first three are: the RandomNAS trained in the GS with Spearman's $\rho$ calculated for 64 architectures randomly samples from top-100% (global), top-60%, top-20% in the GS independently. The other three are from PS: the RandomNAS trained in a PS with 256 architectures sampled from top-100%, top-60%, top-20% the from GS and Spearman's $\rho$ calculated with 64 architectures randomly sampled from the same PS. It shows that the RandomNAS in the GS achieves 0.783 (the blue line) Spearman's $\rho$ with the validation architectures sampled from global. However, it only achieves 0.347 (purple) with the validation architectures sampled from top-20%. On the contrast, the RandomNAS in the PS shows that its Spearman's $\rho$ curve under PS top-100% (orange, 0.785) is similar with the GS top-100%, which is consistent with the Fig. 1b results. However, the Spearman's $\rho$ curve of the PS top-60% (red, 0.687) and top-20% (brown, 0.607) both surpass the GS top-60% (green, 0.530) and GS top-20%. We suspect that such an improvement may be due to the better constructed PS sampled from top performing GS, an insight we will take advantage of in designing our new solution below.

## 2.2 SMALL ARCHITECTURES GAIN LOWER LOSS

We observe another drawback of RandomNAS in that smaller architectures inside the supernet converge faster and tend to gain lower validation loss than the larger ones in the early training phase.

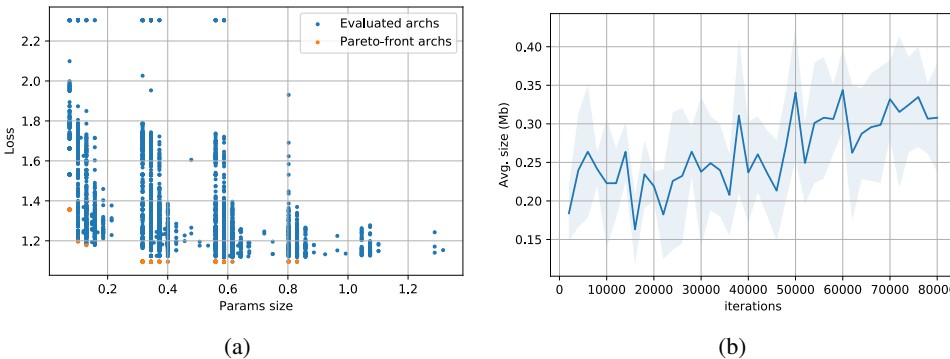

(a)                                          (b)

Figure 2: (a)Validation losses vs. sizes for 4000 architectures after RandomNAS training. (b) Pareto-front architectures' average sizes via RandomNAS training.

Fig. 2a shows the validation losses of 4000 architectures from RandomNAS in the GS after training. Although the architectures with a model size of 1.05Mb achieves higher accuracy (queried from the ground truth in NASBench201), the Pareto front (shown as orange dots) only reaches the 0.83Mb architecture. One possible explanation is that the light-weight operations converge faster. Fig. 2b shows the average size of the validation Pareto front among 320 architectures in each validation interval. We can see that the average sizes of the Pareto-front architectures are small in the early search stage and will increase after more iterations, meaning that the larger architectures gradually begin to converge.

## 3   OUR METHOD

Motivated by the observations in 2.1, we propose a simple yet effective random sampling approach, called EPS, to iteratively improve the architecture quality within the proxy search space. Our proposed solution can distinguish the "better" architectures with higher ranking correlation compared to the existing solutions using global search. Algorithm 1 shows the overall flow of EPS.

**Main Hyper-parameters**. (1) $T\_iter$: the number of iterations for supernet training. (2) $I\_val$: validation interval. (3) $P$: maximum population size, where the population is a proxy search space. (4) $S$: sample size (the number of sampled architectures from population). (5) $M$: the number of mutated architectures. The ablation study on the search hyperparameters will be discussed in Section 4.

**Overall Flow**.

EPS performs three major stages iteratively, as training, validation and evolving, followed by a final architecture selection and retraining. At the beginning, we build a supernet which contains all possible architectures in the search space and initialize $Q\_pop$ and $sample\_set$. RandomInitArch uniformly select a architecture from the GS, and then the architecture is enqueued into the $Q\_pop$. $Q\_pop$ is the proxy search space and $sample\_set$ is randomly sampled from $Q\_pop$ for the tournament selection. (1) **Training**. The supernet is trained by

---

**Algorithm 1:** EPS

Initialize a $supernet$
Initialize an empty architecture population queue, $Q\_pop$
Initialize an empty $history$
**for** $i = 1, 2, \cdots, P$ **do**
  $new\_architecture \leftarrow$
    RandomInitArch($supernet$)
  Enqueue($Q\_pop, new\_architecture$)
**end**
$sample\_set \leftarrow$ RandomSample( $Q\_pop, S$ )
**for** $i = 1, 2, \cdots, T\_iter$ **do**
  $architecture \leftarrow$ RandomSample(
    $sample\_set, 1$ )
  TrainOneBatch($supernet, architecture$)
  **if** $i \% I\_val = 0$ **then**
    **for** $k = 1, 2, \cdots, S$ **do**
      $architecture \leftarrow sample\_set [k]$
      $val\_loss \leftarrow$ ValOneBatch($supernet,$
        $architecture$)
      $architecture.loss \leftarrow val\_loss$
    **end**
    Sort $sample\_set$ by $architecture.loss$ in
      ascending order
    $history.append(sample\_set)$
    **for** $k$ in 1,2...M **do**
      $new\_architecture \leftarrow$
        Mutate($sample\_set[k]$)
      Enqueue( $Q\_pop, new\_architecture$
        )
      Dequeue( $Q\_pop$)
    **end**
    $sample\_set \leftarrow$ RandomSample( $Q\_pop, S$
      )
  **end**
**end**
$selected\_architectures \leftarrow$
  ArchitectureSelection($history, T$)
ArchitectureRetrain($selected\_architectures$)

---

a total of $T\_iter$ iterations. In each iteration, one architecture is sampled from the $sample\_set$ and its weights are updated in the weight-sharing supernet by gradient descent. (2) **Validation**. Every $I\_val$ iterations of training, the architectures inside the $sample\_set$ will be evaluated sequentially. In each validation iteration, one architecture is evaluated on a batch of the randomly sampled validation data, and its validation loss is recorded. We validate the architecture on a single batch instead of the whole validation dataset to find a trade-off between the search time and completeness of validation. (3) **Evolving**. The architecture inside $sample\_set$ are sorted in ascending order according to their latest loss values, and the top-$M$ architectures are mutated and en-queued. Meanwhile, it removes $M$ oldest architecture from $Q\_pop$. The mutation operation is described in Appendix B. (4) **Selection**. After supernet training, we revisit the $history$ of evolution and select all the "winner" architectures which get the lowest loss in the $Q\_pop$ in the each of last $T$ validation intervals. However, if $T$ is large, it still remains numerous number of "winners". Based on the observation from Section 2.2, we propose a simple size regularization: $architecture.loss = architecture.loss + \alpha e^{-\beta size(architecture)/max\_size}$. $max\_size$ is the maximum architecture size in the global search space. The regularization imposes a penalty on small architectures, and the experiment results in Section 4 shows the significant improvement from the regularization. By adding the regularization , we select the one with the lowest regularized loss as the ideal architecture from winners.

# 4 ABLATION STUDY ON NASBENCH-201

We first apply EPS on NASBench-201 (Dong & Yang (2020)) for an ablation study. Following the NASBench-201 settings, we construct a supernet contains the same searchable cells. The number of iterations for training is $T\_iter = 80000$. The main hyper-parameters explored are the maximum population size $P \in \{64, 128, 256\}$, the sample size $S \in \{32, 64\}$, the mutation number $M \in \{1, 4\}$, and the val-

Table 1: Comparison between the EPS, None-aging EA and RSPS on NASBench-201 validation dataset.

| Setting | EPS | Non-aging EA | RSPS |
|---------|-----|--------------|------|
| w/o SR | 90.90±0.10 | 85.61±2.67 | 89.64±0.96 |
| w. SR | **91.50±0.07** | 85.71±2.61 | 90.00±1.08 |

idation interval $I\_val \in \{50, 100, 200\}$. The number of total settings we explore is 36. For the selection, we set $T = 10000 \mod I\_val$. In the first round, the "winner" architectures which get the lowest loss in each of the last $T$ validation interval are selected. If a "winner" dominates several validation intervals, its last loss will be adopted as the criterion. In the second round, the size regularization is added to all the "winner" architectures and we select the one with the lowest regularized loss.

**Size regularization**. We use the validation-training split by NASBench-201 and analyze $\alpha$ and $\beta$ in the size regularization (defined in Sec. 3). We compare the ranking correlation between the architecture's regularized loss and the validation accuracy in the final validation interval of the 36 settings. Each setting runs five times. The average Spearman's $\rho$ (in $36 \times 5$ runs) controlled by $\alpha$ and $\beta$ is shown in Fig. 3a. We explore the $\alpha \in [0, 4]$ and $\beta \in [0, 8]$ with the step 0.1. The average Spearman's $\rho$ reaches 0.796 when $\alpha = 1.4$ and $\beta = 5.2$. Compared to the original Spearman's $\rho$ 0.624, it gains 0.172 improvement with the size regularization.

**Search results**. By adding the regularization, the setting $\{P = 256, S = 64, M = 4, I\_val = 50\}$ achieves the best average validation accuracy of five found architectures. The detailed results of the 36 settings are in the Table 16. We compare the EPS with the RandomNAS (RSPS) and non-aging evolutionary algorithm (EA) in Table 1. RandomNAS follows the same training settings as the EPS and evaluates 4000 architectures on all the validation data after training. Since the RandomNAS in the benchmark used a 256-size data batch for validation of each architecture and only evaluated 100 architectures (please refer to Appendix B, Dong & Yang (2020)), our RandomNAS results higher. EPS surpasses the RandomNAS by a large margin. Our experiment results support the idea that by properly evolving the proxy search space, it can gain a better performance compared to the global RandomNAS. Also, size regularization (SR) alleviates the slower convergence issue of larger architectures and shows the improvements on both EPS and RSPS. For the non-aging EA, architectures with the highest running loss are removed out of $Q\_pop$ instead of the oldest ones. The results show that the non-aging EA is trapped by some poor-performance architectures. The underline reasons are: (1) A young architecture with high loss can be removed before well trained;

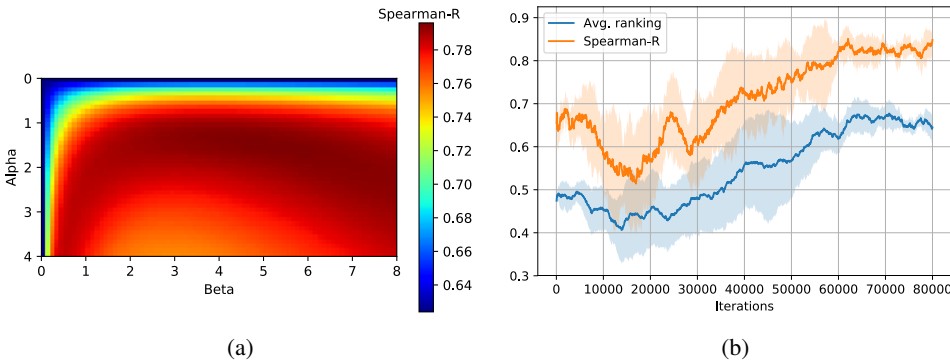

(a)                           (b)

Figure 3: (a) The average ranking correlation of EPS with size regularization on NASBench-201 with $\alpha$ (y-axis) and $\beta$ (x-axis). (b) Average ranking and Spearman's $\rho$ of EPS's proxy search space when training.

(2) An aged architecture survives in the population when it performs well in the early stage and produces many mutations, which may dominate the population and mislead the search direction. We also compare EPS with other state-of-the-art NAS algorithms in Table 2. It shows that EPS delivers a near-optimal results and surpasses NAS works referred in the original Benchmark regarding test accuracy on CIFAR-10 and generalized test accuracy on CIFAR-100 and Imagenet-16. We also plot the average ranking of the proxy search space(population) and the Spearman's $\rho$ of the population in Fig. 3b which shows that EPS successfully improve the average ranking and correlation Spearman's $\rho$ when training.

Table 2: Test accuracy comparisons between EPS and other state-of-the-art NAS works reduplicated in the NASBench-201 paper Li & Talwalkar (2019); Liu et al. (2019b); Dong & Yang (2019b;a); Peng et al. (2019). Our results on three different datasets are respectively listed in the second block.

| Method | Search cost (GPU hours) | Test accuracy(%) | | |
|---|---|---|---|---|
| | | CIFAR-10 | CIFAR-100 | Imagenet16-120 |
| RSPS | 2.2 | 84.07±3.61 | 58.33±4.34 | 26.28±3.09 |
| DARTS-V1 | 3.2 | 54.30±0.00 | 15.61±0.00 | 16.32±0.00 |
| DARTS-V2 | 9.9 | 54.30±0.00 | 15.61±0.00 | 16.32±0.00 |
| GDAS | 8.8 | 93.61±0.09 | 70.61±0.26 | 41.71±0.98 |
| SETN | 9.5 | 87.64±0.00 | 56.87±7.77 | 32.52±0.21 |
| ENAS | 3.9 | 53.89±0.58 | 15.61±0.00 | 14.84±2.10 |
| EPS | 1.8 | **94.34±0.04** | **73.03±0.33** | **46.25±0.29** |
| Optimal | - | 94.37 | 73.49 | 47.31 |

## 5 EXPERIMENTS ON DARTS SEARCH SPACE AND ITS VARIANTS

To further illustrate the generalizability of EPS, we also evaluate it on the DARTS search space and its variants without fine-tuning effort. Following the previous works, we build the network by stacking searchable cells, which contain $N$ nodes as a directed acyclic graph. We first apply EPS in the DARTS image classification search space and 4 different DARTS sub search spaces proposed by Zela et al. (2019). Then we extend the EPS to DARTS language modeling search space, which is completely different from computer vision tasks.

Table 3: Comparisons between EPS and Liu et al. (2019b); Xu et al. (2019) via 5 search runs on CIFAR-10. "*" architectures are reproduced by the work (Xu et al. (2019)).

| Method | Runs | | | | | Avg. err.(%) |
|---|---|---|---|---|---|---|
| | #1 | #2 | #3 | #4 | #5 | |
| DARTSV1* | 2.89 | 3.15 | 2.99 | 3.07 | 3.27 | 3.07±0.13 |
| DARTSV2* | 3.11 | 2.68 | 2.77 | 3.14 | 3.06 | 2.95±0.19 |
| PC-DARTS | 2.72 | 2.67 | 2.57 | 2.75 | 2.64 | 2.67±0.06 |
| EPS | 2.52 | 2.67 | 2.79 | 2.61 | 2.62 | **2.64±0.09** |

## 5.1 DARTS IMAGE CLASSIFICATION SEARCH SPACE

We use the consistent hyper-parameters settings from the NASBench-201 and transfer to the DARTS image search space. Utilizing the random sampling strategy, EPS directly searches on a supernet with 20 cells and 16 number of initial channels with less than 8GB GPU memories consumption. We follow the PC-DARTS evaluations of the searched architectures in five independent search runs, and the results are shown in Table 3. Among the 5 runs, we are able to achieve the lowest average test error compared to DARTS and PC-DARTS. In addition, we make an extensive comparison to more recent NAS works on CIFAR-10 using the best network discovered by EPS within the 5 runs. Also, we do the search follow the same settings on CIFAR-100, and the results are shown in Table 4. It shows that on CIFAR-10/100, EPS is on a par with the recent NAS works and is able to find state-of-the-art architectures.

We design another experiment to compare the EPS to RandomNAS on DARTS search space. The experiment is conducted with 8 cells and 16 number of initial channels for both search and validation. We randomly sample 27 architectures from the last validation interval (after 80,000 training iterations) for EPS. We train the architectures 3 times independently follow the DARTS validation settings. Furthermore, the Spearman's $\rho$ between the EPS(with size regularization) loss and the ground truth is provided. Follow the same EPS training settings, we train a global RandomNAS, and we show both Spearman's $\rho$ in Fig. 4. The randomly sampled 10 architectures average accuracy is plotted by the red line (95.15%). Two observations are made: 1) The architectures sampled from the final proxy search space surpass the random sampling baseline by a large margin (the average accuracy of samples 95.54% vs. 95.15%). 2) EPS delivers 0.68 Spearman's $\rho$ while RandomNAS performs worse (0.41) at distinguishing the difference between them. Note that due to the computational consumption, we could only afford a single repetition for the experiment.

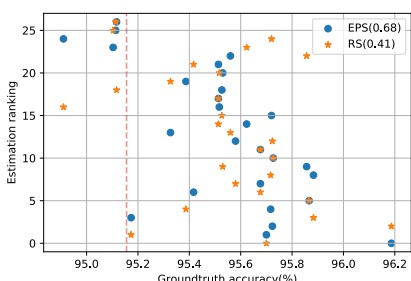

Figure 4: Rankings and ground-truth accuracy of searched architectures from EPS and RandomNAS(RS) on DARTS search space. $\rho$ is calculated by estimated ranking and ground-truth-accuracy ranking(descending). The red line is the average accuracy of random samples.

## 5.2 DARTS IMAGE CLASSIFICATION SUB SEARCH SPACES

In the work (Zela et al. (2019)), it proposed four sub search spaces from DARTS where original DARTS fails. The four search spaces are: S1: A search space contains the most confident 2 operations DARTS predicted on each edge. S2: A search space contains $\{3 \times 3\ SepConv, SkipConnect\}$ on each edge. S3: A search space contains $\{3 \times 3\ SepConv, SkipConnect, None\}$ on each edge.

Table 4: Comparison with the state-of-the-art methods (Peng et al. (2019); Liu et al. (2019b); Dong & Yang (2019b); Xie et al. (2018); Xu et al. (2019); Zela et al. (2019); Lu et al. (2019); Li & Talwalkar (2019); Yang et al. (2020); Zhang et al. (2020)) on CIFAR-10/-100. (Search time may be varied using different GPUs.)

| Method | Test Err.(%) | | Params | Search cost | Search |
| --- | --- | --- | --- | --- | --- |
| | CIFAR-10 | CIFAR-100 | (M) | (GPU days) | method |
| ENAS | 2.89 | 18.91 | 4.6 | 0.45 | RL |
| DARTSV1 | 3.00±0.14 | - | 3.3 | 1.5 | G |
| DARTSV2 | 2.76±0.09 | 17.54* | 3.3 | 4 | G |
| GDAS | 2.82 | 18.13 | 2.5 | 0.17 | G |
| SNAS | 2.85±0.02 | - | 2.8 | 1.5 | G |
| PC-DARTS | 2.57±0.07 | - | 3.6 | 0.1 | G |
| R-DARTS | 2.95±0.21 | 18.01±0.26 | - | 1.5 | G |
| NSGANet | 2.75 | - | 3.3 | 4 | EA |
| RSPS | 2.71 | 17.63* | 4.3 | 0.7 | RS |
| CARS | 2.62 | - | 3.6 | 0.4 | RS&EA |
| RS-NSAS | 2.64(2.50) | 17.56(16.85) | 3.4 | 0.7 | RS |
| EPS | **2.52±0.08(2.41)** | **17.09±0.26(16.90)** | 3.5/3.6 | 0.33 | RS&EA |

Table 5: Comparison with the Liu et al. (2019b), Zela et al. (2019) in DARTS sub search spaces

| Setting | | RSPS | DARTS | DARTS-ES | DARTS-ADA | EPS |
|---|---|---|---|---|---|---|
| | S1 | 3.17±0.15 | 4.66±0.71 | 3.05±0.07 | 3.03±0.08 | **2.84±0.16** |
| CIFAR-10 | S2 | 3.46±0.15 | 4.42±0.40 | 3.41±0.14 | 3.59±0.3 | **3.21±0.06** |
| | S3 | 2.92±0.04 | 4.12±0.85 | 3.71±1.14 | 2.99±0.34 | **2.61±0.05** |
| | S4 | 89.39±0.84 | 6.95±0.18 | 4.17±0.21 | 3.89±0.67 | **3.82±0.15** |
| | S1 | 25.81±0.39 | 29.93±0.41 | 28.90±0.81 | 24.94±0.81 | **23.85±1.27** |
| CIFAR-100 | S2 | 22.88±0.16 | 28.75±0.92 | 24.68±1.43 | 26.88±1.11 | **21.92±0.27** |
| | S3 | 24.58±0.61 | 29.01±0.24 | 26.99±1.79 | 24.55±0.63 | **22.38±0.61** |
| | S4 | 30.01±1.52 | 24.77±1.51 | 23.90±2.01 | **23.66±0.90** | 23.93±0.67 |
| | S1 | 2.64±0.09 | 9.88±5.50 | 2.80±0.09 | 2.59±0.07 | **2.49±0.04** |
| SVHN | S2 | 2.57±0.04 | 3.69±0.12 | 2.68±0.18 | 2.79±0.22 | **2.56±0.09** |
| | S3 | 2.89±0.09 | 4.00±1.01 | 2.78±0.29 | 2.58±0.07 | **2.53±0.05** |
| | S4 | 3.42±0.04 | 2.90±0.02 | 2.55±0.15 | **2.52±0.06** | 2.78±0.11 |

S4: A search space contains $\{3 \times 3 \, SepConv, \, Noise\}$ on each edge. Also, we noticed that RandomNAS tend to fail on the S4 which suggests the method is difficult to distinguishes the $Noise$ operation from $3 \times 3 \, SepConv$.

Here, we use similar settings for the EPS. We use the DARTS maximum architecture's size as $max\_size$ for the size regularization and set the size for $Noise$ simply as negative $3 \times 3 \, SepConv$ size since it disrupts the information on its edge. Also, we notice that $None$ operation is as a placeholder for the DNAS training and not in the final architecture. Thus, S3 is the same as S2 for EPS. The results are shown in Table 5. For each setting, we report the 3 found architectures mean and std follow Table 6 in Zela et al. (2019). EPS delivers better performance on the most of them. Even in the S4 where RandomNAS is fragile, EPS still performs robustly and achieves the better performance on 2 settings out of three compared to DARTS-ADA. EPS even find an architectures on CIFAR-10, S3 with 2.55±0.05 testing error, which is the state-of-the-art in the whole search space (In Table 12,S3#1).

## 5.3 DARTS PTB SEARCH SPACE

We also evaluate EPS on the Penn Treebank (PTB) dataset, which targets natural language processing tasks and is completely different from computer vision tasks. The performance evaluation usually uses the perplexity score, the lower, the better. We use the DARTS PTB search space for the recurrent cells. During the recurrent cell search, the hidden and embedding sizes are both set to be 850, and the hyperparameters settings are similar to the previous tasks. Please refer to the Appendix B.4 for the details. Since operations in the search space are weight-free, we use a simple selection strategy

Table 6: Comparison with other state-of-the-art methods on PTB.

| Method | Perplexity | | Search | Search |
|---|---|---|---|---|
| | Valid | Test | cost (GPU days) | method |
| ENAS | 60.8 | 58.6 | 0.5 | RL |
| DARTS-V1 | 60.2 | 57.6 | 0.5 | G |
| DARTS-V2 | 58.1 | 55.7 | 1 | G |
| GDAS | 59.8 | 57.5 | 0.4 | G |
| G-NSAS | 59.74 | 57.24 | 0.5 | G |
| RSPS | 57.8 | 55.5 | 0.25 | RS |
| RS-NSAS | 59.22 | 56.84 | 0.62 | RS |
| EPS | 58.4 | 56.27 | 0.5 | RS&EA |

by training the last 10000 iteration's "winner" architectures for two epochs and evaluate the top-5 architectures for 20 epochs. So the training and selection take 8+4 hours (0.5 GPU day). After the search, we use the same settings as in DARTS to train the best networks for 3600 epochs. The best model discovered by EPS achieves a validation perplexity of 58.4 and a test perplexity of 56.27, which is on a par with the state-of-the-art approaches. The full comparisons are shown in Table 6.

## 6 CONCLUSION

We showed that RandomNAS had performed poorly in ranking good architectures and suffered from the lower validation loss for small architectures. Based on the proxy search space, we observed the ranking correlation between the prediction and the ground truth among good architectures could be improved if using a proxy search space consists of good architectures. Hence, we proposed an efficient way to Evolve the Proxy Search Space (EPS). We also designed a simple size regularization to help RandomNAS-based algorithm jump out of the small architecture traps. In the NASBench-201 experiments, EPS delivered a near-optimal solution and surpassed the existing methods. In the

extensive experiments on DARTS search spaces and its variants, EPS outperformed or was tied with the majority of recent NAS works. We believe such observations and insights as presented in this paper can be useful to the community for the future NAS research with better interpretability.

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

## A  BACKGROUND AND RELATED WORK

**Neural Architecture Search**. Neural Architecture Search has demonstrated its capability of automatic DNN architecture design and generated DNNs even better than handcrafted ones in several tasks like object detection (Chen et al. (2019b); Peng et al. (2019)), segmentation (Liu et al. (2019a)) and disparity estimation (Saikia et al. (2019)). Early works, such as Zoph & Le (2017); Real et al. (2019) use reinforcement learning or evolutionary algorithm to deliver optimal network structures in discrete search spaces but spend tremendous time on architecture searching and network training. Recently, more efficient methods, such as one-shot and weight-sharing NAS, are widely adopted by the community to reduce the network search time. Previously published works (Liu et al. (2019b); Cai et al. (2018); Xie et al. (2018); Dong & Yang (2019b); Xu et al. (2019); Chen et al. (2019a)) start to use differentiable representation for the search space and focus on optimizing the architecture search speed. Other works (Li & Talwalkar (2019); Chen et al. (2019b); Zhang et al. (2020); Guo et al. (2019); Bender (2019); Yang et al. (2020)) adopt discrete and random sampling strategies to explore architectures in a weight-sharing supernet. Recently, we have seen a rising concern about the evaluation and effectiveness of NAS (Yang et al. (2019); Yu et al. (2019); Zela et al. (2019)). Therefore, researchers started to launch benchmarks to provide fair and efficient NAS evaluations (Ying et al. (2019); Zela et al. (2020); Dong & Yang (2020)).

**Random Sampling NAS**. As one of the major approaches, random sampling NAS has been widely studied in recent years. Bender et al. (2018) proposed a one-shot network training strategy with the dropout of operations, which can be considered as an extreme case of the random sampling. Later, Li & Talwalkar (2019) explored the reproducibility issue of existing works and observed that the Random Sampling strategy creates a strong baseline on image classification and language modeling. Guo et al. (2019) proposed a strategy to combine the uniformly random sampling training and post-training evaluation by an evolutionary algorithm to deliver more efficient solutions for image classification. To improve the search latency, Yang et al. (2020) used a modified NSGA-III to generate the Pareto frontier with a fast search time; while Zhang et al. (2020) proposed a general approach to maximize the search diversity and enhance the ranking correlation for both DNAS and RandomNAS. Also, RandomNAS has shown its strength in developing hardware-efficient network designs (Cai et al. (2019); Yu et al. (2020)) Although most of the works have achieved the state-of-the-art performance in their domains, few of them have investigated the ranking correlation between the prediction of architectures by RandomNAS and the ground truth, which is the keystone for the success of the RandomNAS as a ranking estimator.

**Differences between EPS and other works**. Although we use the tournament selection EA and aging mechanism for the EPS, which is also adopted by Real et al. (2019) (named regularized evolutionary algorithm), the motivation is different. We find the effectiveness of such a method for updating the proxy search space in a weight-sharing supernet while Real et al. (2019) took it as a start-from-scratch-training selection for independent architectures. Thus, EPS searching time is close to the time to train an architecture from the search space. However, Real et al. (2019) takes more than 3,000 GPU days to find an architecture that leads to a 9,000X consumption than ours. Also, we notice a recent work, the CARS(Yang et al. (2020)) uses a modified NSGA-III for the selection in the training of a supernet. The main differences are 1. EPS motivates by several observations of RandomNAS, which is very different from the CARS. 2. EPS uses a single object EA for minimizing the validation losses while CARS uses an NSGA-III based algorithm for two conflicting objectives (potentially): maximizing the performance while minimizing the architectures' parameter size. 3. EPS updates a single architecture at one time while CARS updates multiple architectures and leads to higher GPU memories consumption. 4. EPS uses a batch of data for validation of each architecture while CARS uses the whole validation dataset for an architecture. Overall, EPS shows a higher accuracy on DARTS search space compared to CARS, and since CARS isn't open-sourced we are unable to do the extensive experiments for it.

## B  DETAILED SETTINGS

### B.1  NASBench-201

Our works are done on PyTorch(Paszke et al. (2019)). Works like Yu et al. (2018); Liu et al. (2019b) point out the global moving average of the batch normalization leads to fluctuation among the

weight-sharing architectures. In the EPS, we disable the learnable affine parameters in all the batch normalization layers. We strictly follow the NASBench-201 settings to split the CIFAR-10 training dataset into train/validation two parts(1:1). We use the SGD with the momentum as 0.9. The initial learning rate is 0.1. The weight decay is $4 \times 10^{-5}$. The scheduler is the cosine annealing from 1 to 0. The training batch size is 128 and the validation batch size is 250(1% of the training data). The loss function is a label-smoothing(Szegedy et al. (2016)) cross entropy loss with $\alpha_{smooth} = 0.1$ which contains more information of the prediction distribution than the one-hot criterion. The evolutionary algorithm only mutates the cells. The rule for mutation is: The current operation on the edge have $P\_opwise = 0.5$ as the probability to be mutated to a new operation(include itself) on the same edge. For the data augmentation we only use the random cropping with padding 4, the random horizontal flipping and image normalization based on the training dataset statistics.

### B.2 DARTS Image Classification Search Space

The settings are the same as B.1 except the mutation. Since the DARTS search space is larger than NASBench-201($10^{18}$ vs. 15,625), we adopt a simple yet conservative mutation rule. 1. The current operation on the edge have $P\_opwise = 0.2$ as the probability to be mutated to a new operation(include itself) on the same edge. 2. If a node has the unconnected predecessors, one of its edges have $P\_opwise = 0.1$ as the probability to switch to an edge linked the unconnected predecessors, and the operation on the new edge will be chosen randomly. We apply this settings through all DARTS-search-space series. For the search supernet, we choose 16 initial channels and 20 stacked cells. For training architectures from the scratch, we strictly follow Liu et al. (2019b) settings to generate a fairly comparable results.

### B.3 DARTS Image Classification Sub Search Space

Follow Zela et al. (2019); Liu et al. (2019b), for CIFAR-100 and SVHN we use 16 number of initial channels and 8 cells when training architectures from scratch. For CIFAR-10 S1 and S3, we use 36 initial channels and 20 stacked cell. For S2 and S4 we use 16 initial channels and 20 stacked cell. During the EPS searching, we use 16 number of initial channels and 8 cells for the supernet of CIFAR-100 and SVHN. we use 16 number of initial channels and 20 cells for the supernet of CIFAR-10.

### B.4 DARTS PTB Search Space

We use the settings for search as Liu et al. (2019b) except: 1. We choose the embedding and hidden dimension of 850 for the RNN. 2. We run the experiment for 80,000 iterations with 128 training batch size and uses the $\{P$:256, $S$:64, $M$:4, $I\_val$:50$\}$ for EA. 3. The gradient clip is changed to 0.1. 4. We only use 1/4 validation dataset for the architecture validation. The settings for training an architecture from the scratch is exactly the same.

## C Additional Experiment

### C.1 Optional approaches on NASBench-201

Here we discuss two optional methods and evaluated them on NASBench-201. The first one is the progressive search. The algorithm is shown in Alg. 2. The supernet is initialized with 3 searchable cells ($N = 1$ and total searchable cells are $3N$). Please refer to the Figure 1 in the NASBench-201 paper(Dong & Yang (2020) for the defination of $N$). We gradually increase $N$ by 1 in each growth interval ($G_val = 15000$ iterations) until $N_max = 5$. The new cells are randomly initialized and stacked into the supernet. Other training settings are adopted from EPS. The second one is the EPS with multi layer perceptron(MLP). The algorithm is shown in Alg. 3. The idea is to use the MLP learn the losses of architectures in the current $Q\_pop$ and to predict architectures in the GS. We use a 3-layer MLP with 100 as the embedding size. The input is the indexes of the operations on 6 edges. In MLPTrain, we use the architectures evaluated in current $Q\_pop$ as the input and their losses as the label to finetune the MLP for 20 epochs. In the MLPPredict, we use the current MLP to predict losses of 256 architectures randomly generated. Then it return the best predicted architecture. The five-run experiment results on NASBench-201 are shown in the Table 7. Original EPS outperforms

other two methods on CIFAR-10. Since Progressive EPS speeds up 1.5x, we considered it as a good trade-off between the performance and the speed.

---

**Algorithm 2:** Progressive EPS

---

Initialize a $supernet$ with $N = 1$
Initialize an empty architecture population queue, $Q\_pop$
Initialize an empty $history$
**for** $i = 1, 2, \cdots, P$ **do**
    $new\_architecture \leftarrow$ RandomInitArch($supernet$)
    Enqueue($Q\_pop, new\_architecture$)
**end**
$sample\_set \leftarrow$ RandomSample( $Q\_pop, S$ )
**for** $i = 1, 2, \cdots, T\_iter$ **do**
    $architecture \leftarrow$ RandomSample( $sample\_set, 1$ )
    TrainOneBatch($supernet, architecture$)
    **if** $i \% I\_val = 0$ **then**
        **for** $k = 1, 2, \cdots, S$ **do**
            $architecture \leftarrow sample\_set[k]$
            $val\_loss \leftarrow$ ValOneBatch($supernet, architecture$)
            $architecture.loss \leftarrow val\_loss$
        **end**
        Sort $sample\_set$ by $architecture.loss$ in ascending order
        $history.append(sample\_set)$
        **for** $k$ in 1,2...M **do**
            $new\_architecture \leftarrow$ Mutate($sample\_set[k]$)
            Enqueue( $Q\_pop, new\_architecture$ )
            Dequeue( $Q\_pop$)
        **end**
        $sample\_set \leftarrow$ RandomSample( $Q\_pop, S$ )
        **if** $i \% G\_val = 0$ **AND** $N < N_{max}$ **then**
            $N = N + 1$
        **end**
    **end**
**end**
$selected\_architectures \leftarrow$ ArchitectureSelection($history$,$T$)
ArchitectureRetrain($selected\_architectures$)

---

**Algorithm 3:** EPS with MLP

---

Initialize a $supernet$
Initialize an empty architecture population queue, $Q\_pop$
Initialize an 3-layer $MLP$
Initialize an empty $history$
**for** $i = 1, 2, \cdots, P$ **do**
    $new\_architecture \leftarrow$ RandomInitArch($supernet$)
    Enqueue($Q\_pop, new\_architecture$)
**end**
$sample\_set \leftarrow$ RandomSample( $Q\_pop, S$ )
**for** $i = 1, 2, \cdots, T\_iter$ **do**
    $architecture \leftarrow$ RandomSample( $sample\_set, 1$ )
    TrainOneBatch($supernet, architecture$)
    **if** $i \% I\_val = 0$ **then**
        **for** $k = 1, 2, \cdots, S$ **do**
            $architecture \leftarrow sample\_set[k]$
            $val\_loss \leftarrow$ ValOneBatch($supernet, architecture$)
            $architecture.loss \leftarrow val\_loss$
        **end**
        MLPTrain($MLP$,$Q\_pop$)
        Sort $sample\_set$ by $architecture.loss$ in ascending order
        $history.append(sample\_set)$
        **for** $k$ in 1,2...M **do**
            $new\_architecture \leftarrow$ MLPPredict($MLP$)
            Enqueue( $Q\_pop, new\_architecture$ )
            Dequeue( $Q\_pop$)
        **end**
        $sample\_set \leftarrow$ RandomSample( $Q\_pop, S$ )
    **end**
**end**
$selected\_architectures \leftarrow$ ArchitectureSelection($history$,$T$)
ArchitectureRetrain($selected\_architectures$)

---

Table 7: Comparison between EPS and two optional approaches on NASBench-201

| Method | Valid acc. | Test acc. | GPU hrs. |
|---|---|---|---|
| Progressive EPS | 91.33 | 94.00 | 1.2 |
| EPS with MLP | 89.17 | 92.57 | 2.0 |
| EPS | **91.50** | **94.34** | 1.8 |

## C.2 4-RUN TRIAL ON DARTS PTB SEARCH SPACE

Similar to the work Li & Talwalkar (2019), we conduct an experiment to run EPS 4 times and compare the 300-epoch validation perplexity of the selected architectures. The results are shown in Table 8.

Table 8: EPS is ran for 4 times with different random seeds. The best of these architectures for each run is then trained from scratch for 300 epochs.

| Method | Run1 | Run2 | Run3 | Run4 | Avg. perplexity |
|---|---|---|---|---|---|
| DARTS | 67.3 | 66.3 | 63.4 | 63.4 | 65.1 |
| RSPS | 66.3 | 64.6 | 64.1 | 63.8 | 64.7 |
| EPS | 65.10 | **63.74** | 64.72 | 64.77 | **64.58** |

# D    MORE DETAILS

Table 9: The GPU benchmark of the EPS in different search spaces on one Titan V GPU.

| Settings | Avg. load | | Max. load | | |
|---|---|---|---|---|---|
| | Train speed (batch/sec.) | Val. speed (batch/sec.) | Train speed (batch/sec.) | Val. speed (batch/sec.) | Mem. (GB) |
| NASBench-201 | 26.11 | 45.79 | 15.55 | 24.09 | 2.27 |
| DARTS(8channels,16cells) | 10.07 | 27.25 | 8.42 | 17.57 | 3.88 |
| DARTS(20channels,16cells) | 3.91 | 11.51 | 3.68 | 6.83 | 8.01 |
| DARTS-PTB | 6.83 | 6.84 | N/A | N/A | 3.79 |

Table 10: Architectures searched on NASBench-201.

| Setting | Architecture |
| --- | --- |
| #1 | `|nor_conv_3x3~0|+|nor_conv_3x3~0|nor_conv_3x3~1|+|skip_connect~0|nor_conv_3x3~1|nor_conv_3x3~2|` |
| #2 | `|nor_conv_3x3~0|+|nor_conv_3x3~0|nor_conv_1x1~1|+|skip_connect~0|nor_conv_3x3~1|nor_conv_3x3~2|` |
| #3 | `|nor_conv_3x3~0|+|nor_conv_3x3~0|nor_conv_3x3~1|+|skip_connect~0|nor_conv_1x1~1|nor_conv_3x3~2|` |
| #4 | `|nor_conv_3x3~0|+|nor_conv_3x3~0|nor_conv_1x1~1|+|skip_connect~0|nor_conv_3x3~1|nor_conv_3x3~2|` |
| #5 | `|nor_conv_3x3~0|+|nor_conv_3x3~0|nor_conv_3x3~1|+|skip_connect~0|nor_conv_3x3~1|nor_conv_1x1~2|` |

Table 11: Architectures searched on DARTS image classification search space.

| Setting | | Architecture |
|---|---|---|
| CIFAR-10 | #1 | `Genotype(`**`normal`**`=[('max_pool_3x3', 0), ('sep_conv_5x5', 1), ('skip_conn ect', 0), ('sep_conv_3x3', 1), ('sep_conv_3x3', 0), ('sep_conv_5x5', 1 ), ('dil_conv_5x5', 1), ('dil_conv_5x5', 2)], `**`normal_concat`**`=`**`range`**`(2, 6 ), `**`reduce`**`=[('sep_conv_5x5', 0), ('dil_conv_3x3', 1), ('avg_pool_3x3', 0), ('dil_conv_3x3', 2), ('sep_conv_3x3', 1), ('avg_pool_3x3', 3), ('a vg_pool_3x3', 1), ('avg_pool_3x3', 2)], `**`reduce_concat`**`=`**`range`**`(2, 6))` |
| | #2 | `Genotype(`**`normal`**`=[('sep_conv_3x3', 0), ('skip_connect', 1), ('sep_conv_ 3x3', 1), ('dil_conv_3x3', 2), ('sep_conv_3x3', 0), ('sep_conv_5x5', 2 ), ('max_pool_3x3', 0), ('sep_conv_5x5', 1)], `**`normal_concat`**`=`**`range`**`(2, 6 ), `**`reduce`**`=[('sep_conv_5x5', 0), ('dil_conv_3x3', 1), ('avg_pool_3x3', 0), ('sep_conv_3x3', 2), ('avg_pool_3x3', 0), ('sep_conv_5x5', 1), ('a vg_pool_3x3', 0), ('skip_connect', 1)], `**`reduce_concat`**`=`**`range`**`(2, 6))` |
| | #3 | `Genotype(`**`normal`**`=[('skip_connect', 0), ('skip_connect', 1), ('sep_conv_ 3x3', 0), ('dil_conv_5x5', 2), ('sep_conv_3x3', 0), ('sep_conv_3x3', 2 ), ('dil_conv_5x5', 0), ('sep_conv_5x5', 1)], `**`normal_concat`**`=`**`range`**`(2, 6 ), `**`reduce`**`=[('sep_conv_5x5', 0), ('dil_conv_5x5', 1), ('skip_connect', 0), ('dil_conv_5x5', 1), ('avg_pool_3x3', 2), ('sep_conv_5x5', 3), ('s kip_connect', 2), ('max_pool_3x3', 3)], `**`reduce_concat`**`=`**`range`**`(2, 6))` |
| | #4 | `Genotype(`**`normal`**`=[('skip_connect', 0), ('dil_conv_3x3', 1), ('dil_conv_ 5x5', 0), ('sep_conv_5x5', 1), ('sep_conv_3x3', 0), ('sep_conv_5x5', 1 ), ('skip_connect', 1), ('sep_conv_3x3', 3)], `**`normal_concat`**`=`**`range`**`(2, 6 ), `**`reduce`**`=[('sep_conv_3x3', 0), ('max_pool_3x3', 1), ('skip_connect', 0), ('max_pool_3x3', 1), ('dil_conv_5x5', 1), ('max_pool_3x3', 2), ('a vg_pool_3x3', 0), ('dil_conv_3x3', 2)], `**`reduce_concat`**`=`**`range`**`(2, 6))` |
| | #5 | `Genotype(`**`normal`**`=[('skip_connect', 0), ('sep_conv_3x3', 1), ('sep_conv_ 5x5', 0), ('sep_conv_3x3', 1), ('skip_connect', 2), ('sep_conv_5x5', 3 ), ('dil_conv_3x3', 0), ('dil_conv_5x5', 1)], `**`normal_concat`**`=`**`range`**`(2, 6 ), `**`reduce`**`=[('dil_conv_3x3', 0), ('sep_conv_3x3', 1), ('avg_pool_3x3', 0), ('skip_connect', 2), ('sep_conv_3x3', 1), ('sep_conv_5x5', 3), ('s ep_conv_3x3', 1), ('sep_conv_3x3', 3)], `**`reduce_concat`**`=`**`range`**`(2, 6))` |
| CIFAR-100 | #1 | `Genotype(`**`normal`**`=[('skip_connect', 0), ('skip_connect', 1), ('dil_conv_ 3x3', 1), ('sep_conv_5x5', 2), ('dil_conv_5x5', 0), ('dil_conv_5x5', 1 ), ('sep_conv_3x3', 0), ('sep_conv_5x5', 1)], `**`normal_concat`**`=`**`range`**`(2, 6 ), `**`reduce`**`=[('dil_conv_3x3', 0), ('sep_conv_3x3', 1), ('sep_conv_3x3', 1), ('dil_conv_3x3', 2), ('sep_conv_3x3', 1), ('sep_conv_5x5', 3), ('s kip_connect', 0), ('skip_connect', 3)], `**`reduce_concat`**`=`**`range`**`(2, 6))` |
| | #2 | `Genotype(`**`normal`**`=[('dil_conv_3x3', 0), ('skip_connect', 1), ('dil_conv_ 3x3', 0), ('sep_conv_5x5', 1), ('skip_connect', 0), ('dil_conv_5x5', 2 ), ('sep_conv_5x5', 1), ('sep_conv_5x5', 2)], `**`normal_concat`**`=`**`range`**`(2, 6 ), `**`reduce`**`=[('sep_conv_5x5', 0), ('sep_conv_3x3', 1), ('avg_pool_3x3', 0), ('sep_conv_5x5', 1), ('sep_conv_3x3', 1), ('skip_connect', 2), ('s kip_connect', 2), ('sep_conv_3x3', 4)], `**`reduce_concat`**`=`**`range`**`(2, 6))` |
| | #3 | `Genotype(`**`normal`**`=[('skip_connect', 0), ('skip_connect', 1), ('dil_conv_ 5x5', 0), ('dil_conv_5x5', 2), ('dil_conv_5x5', 0), ('sep_conv_3x3', 2 ), ('sep_conv_5x5', 0), ('sep_conv_3x3', 1)], `**`normal_concat`**`=`**`range`**`(2, 6 ), `**`reduce`**`=[('dil_conv_3x3', 0), ('sep_conv_5x5', 1), ('skip_connect', 0), ('sep_conv_5x5', 1), ('sep_conv_3x3', 0), ('dil_conv_3x3', 3), ('m ax_pool_3x3', 0), ('max_pool_3x3', 1)], `**`reduce_concat`**`=`**`range`**`(2, 6))` |
| | #4 | `Genotype(`**`normal`**`=[('dil_conv_3x3', 0), ('skip_connect', 1), ('dil_conv_ 5x5', 0), ('sep_conv_5x5', 2), ('sep_conv_5x5', 0), ('sep_conv_5x5', 1 ), ('skip_connect', 0), ('dil_conv_5x5', 1)], `**`normal_concat`**`=`**`range`**`(2, 6 ), `**`reduce`**`=[('skip_connect', 0), ('dil_conv_5x5', 1), ('sep_conv_5x5', 0), ('max_pool_3x3', 1), ('sep_conv_5x5', 2), ('skip_connect', 3), ('s ep_conv_3x3', 0), ('sep_conv_3x3', 1)], `**`reduce_concat`**`=`**`range`**`(2, 6))` |
| | #5 | `Genotype(`**`normal`**`=[('skip_connect', 0), ('skip_connect', 1), ('sep_conv_ 3x3', 0), ('sep_conv_5x5', 2), ('sep_conv_5x5', 0), ('dil_conv_5x5', 2 ), ('sep_conv_5x5', 2), ('dil_conv_5x5', 3)], `**`normal_concat`**`=`**`range`**`(2, 6 ), `**`reduce`**`=[('avg_pool_3x3', 0), ('sep_conv_3x3', 1), ('max_pool_3x3', 0), ('avg_pool_3x3', 1), ('sep_conv_5x5', 0), ('dil_conv_3x3', 1), ('a vg_pool_3x3', 2), ('sep_conv_5x5', 3)], `**`reduce_concat`**`=`**`range`**`(2, 6))` |

Table 12: Architectures searched on DARTS image classification sub search space (CIFAR-10).

| Setting | | Architecture |
|---|---|---|
| S1 | #1 | **Genotype**(**normal**=[('dil_conv_3x3', 0), ('skip_connect', 1), ('dil_conv_5x5', 0), ('skip_connect', 2), ('skip_connect', 0), ('sep_conv_3x3', 2), ('dil_conv_5x5', 3), ('dil_conv_5x5', 4)], **normal_concat**=**range**(2, 6), **reduce**=[('max_pool_3x3', 0), ('max_pool_3x3', 1), ('max_pool_3x3', 0), ('dil_conv_5x5', 2), ('avg_pool_3x3', 1), ('dil_conv_5x5', 3), ('max_pool_3x3', 0), ('skip_connect', 2)], **reduce_concat**=**range**(2, 6)) |
| | #2 | **Genotype**(**normal**=[('skip_connect', 0), ('skip_connect', 1), ('dil_conv_5x5', 0), ('skip_connect', 2), ('sep_conv_3x3', 1), ('skip_connect', 3), ('sep_conv_3x3', 0), ('dil_conv_3x3', 2)], **normal_concat**=**range**(2, 6), **reduce**=[('max_pool_3x3', 0), ('max_pool_3x3', 1), ('dil_conv_5x5', 2), ('sep_conv_3x3', 1), ('dil_conv_3x3', 2), ('dil_conv_5x5', 3), ('dil_conv_5x5', 4)], **reduce_concat**=**range**(2, 6)) |
| | #3 | **Genotype**(**normal**=[('dil_conv_3x3', 0), ('dil_conv_5x5', 1), ('sep_conv_3x3', 1), ('dil_conv_3x3', 2), ('skip_connect', 0), ('sep_conv_3x3', 2), ('sep_conv_3x3', 0), ('dil_conv_3x3', 2)], **normal_concat**=**range**(2, 6), **reduce**=[('max_pool_3x3', 0), ('max_pool_3x3', 1), ('avg_pool_3x3', 0), ('dil_conv_5x5', 2), ('max_pool_3x3', 0), ('skip_connect', 2), ('max_pool_3x3', 0), ('max_pool_3x3', 1)], **reduce_concat**=**range**(2, 6)) |
| S2 | #1 | **Genotype**(**normal**=[('skip_connect', 0), ('sep_conv_3x3', 1), ('sep_conv_3x3', 0), ('skip_connect', 1), ('sep_conv_3x3', 0), ('sep_conv_3x3', 3), ('sep_conv_3x3', 2), ('sep_conv_3x3', 4)], **normal_concat**=**range**(2, 6), **reduce**=[('sep_conv_3x3', 0), ('sep_conv_3x3', 1), ('skip_connect', 0), ('sep_conv_3x3', 1), ('sep_conv_3x3', 2), ('sep_conv_3x3', 3), ('sep_conv_3x3', 1), ('skip_connect', 4)], **reduce_concat**=**range**(2, 6)) |
| | #2 | **Genotype**(**normal**=[('skip_connect', 0), ('sep_conv_3x3', 1), ('skip_connect', 1), ('sep_conv_3x3', 2), ('sep_conv_3x3', 0), ('sep_conv_3x3', 2), ('sep_conv_3x3', 0), ('sep_conv_3x3', 1)], **normal_concat**=**range**(2, 6), **reduce**=[('sep_conv_3x3', 0), ('skip_connect', 1), ('sep_conv_3x3', 0), ('sep_conv_3x3', 2), ('skip_connect', 0), ('sep_conv_3x3', 3), ('skip_connect', 0), ('skip_connect', 4)], **reduce_concat**=**range**(2, 6)) |
| | #3 | **Genotype**(**normal**=[('sep_conv_3x3', 0), ('sep_conv_3x3', 1), ('sep_conv_3x3', 0), ('sep_conv_3x3', 1), ('sep_conv_3x3', 0), ('skip_connect', 2), ('skip_connect', 0), ('sep_conv_3x3', 1)], **normal_concat**=**range**(2, 6), **reduce**=[('skip_connect', 0), ('skip_connect', 1), ('skip_connect', 1), ('sep_conv_3x3', 2), ('sep_conv_3x3', 3), ('sep_conv_3x3', 2), ('sep_conv_3x3', 3)], **reduce_concat**=**range**(2, 6)) |
| S3 | #1 | **Genotype**(**normal**=[('sep_conv_3x3', 0), ('sep_conv_3x3', 1), ('sep_conv_3x3', 0), ('skip_connect', 1), ('sep_conv_3x3', 0), ('sep_conv_3x3', 1), ('sep_conv_3x3', 0), ('sep_conv_3x3', 1)], **normal_concat**=**range**(2, 6), **reduce**=[('skip_connect', 0), ('skip_connect', 1), ('skip_connect', 2), ('skip_connect', 0), ('sep_conv_3x3', 2), ('sep_conv_3x3', 0), ('sep_conv_3x3', 1)], **reduce_concat**=**range**(2, 6)) |
| | #2 | **Genotype**(**normal**=[('skip_connect', 0), ('sep_conv_3x3', 1), ('skip_connect', 1), ('skip_connect', 2), ('sep_conv_3x3', 0), ('sep_conv_3x3', 1), ('sep_conv_3x3', 0), ('sep_conv_3x3', 1)], **normal_concat**=**range**(2, 6), **reduce**=[('sep_conv_3x3', 0), ('skip_connect', 1), ('sep_conv_3x3', 0), ('sep_conv_3x3', 2), ('skip_connect', 1), ('sep_conv_3x3', 3), ('sep_conv_3x3', 0), ('skip_connect', 3)], **reduce_concat**=**range**(2, 6)) |
| | #3 | **Genotype**(**normal**=[('sep_conv_3x3', 0), ('sep_conv_3x3', 1), ('skip_connect', 1), ('sep_conv_3x3', 1), ('sep_conv_3x3', 0), ('sep_conv_3x3', 1), ('sep_conv_3x3', 0), ('sep_conv_3x3', 2)], **normal_concat**=**range**(2, 6), **reduce**=[('sep_conv_3x3', 0), ('sep_conv_3x3', 1), ('skip_connect', 1), ('sep_conv_3x3', 2), ('skip_connect', 1), ('skip_connect', 3), ('sep_conv_3x3', 1), ('sep_conv_3x3', 4)], **reduce_concat**=**range**(2, 6)) |
| S4 | #1 | **Genotype**(**normal**=[('sep_conv_3x3', 0), ('sep_conv_3x3', 1), ('sep_conv_3x3', 0), ('sep_conv_3x3', 1), ('sep_conv_3x3', 2), ('sep_conv_3x3', 3), ('sep_conv_3x3', 0), ('sep_conv_3x3', 4)], **normal_concat**=**range**(2, 6), **reduce**=[('sep_conv_3x3', 0), ('sep_conv_3x3', 1), ('sep_conv_3x3', 0), ('noise', 1), ('sep_conv_3x3', 2), ('sep_conv_3x3', 3), ('sep_conv_3x3', 0), ('sep_conv_3x3', 3)], **reduce_concat**=**range**(2, 6)) |
| | #2 | **Genotype**(**normal**=[('sep_conv_3x3', 0), ('sep_conv_3x3', 1), ('sep_conv_3x3', 0), ('sep_conv_3x3', 1), ('sep_conv_3x3', 0), ('noise', 3), ('sep_conv_3x3', 0), ('sep_conv_3x3', 2)], **normal_concat**=**range**(2, 6), **reduce**=[('sep_conv_3x3', 0), ('noise', 1), ('noise', 0), ('sep_conv_3x3', 1), ('noise', 1), ('sep_conv_3x3', 3), ('sep_conv_3x3', 0), ('sep_conv_3x3', 1)], **reduce_concat**=**range**(2, 6)) |
| | #3 | **Genotype**(**normal**=[('sep_conv_3x3', 0), ('sep_conv_3x3', 1), ('sep_conv_3x3', 0), ('sep_conv_3x3', 2), ('sep_conv_3x3', 0), ('sep_conv_3x3', 1), ('sep_conv_3x3', 0), ('sep_conv_3x3', 3)], **normal_concat**=**range**(2, 6), **reduce**=[('noise', 0), ('noise', 1), ('sep_conv_3x3', 0), ('sep_conv_3x3', 1), ('noise', 0), ('sep_conv_3x3', 1), ('noise', 1), ('sep_conv_3x3', 2)], **reduce_concat**=**range**(2, 6)) |

Table 13: Architectures searched on DARTS image classification sub search space (CIFAR-100).

| Setting | | Architecture |
|---|---|---|
| S1 | #1 | **Genotype**(**normal**=[('skip_connect', 0), ('dil_conv_5x5', 1), ('dil_conv_5x5', 0), ('sep_conv_3x3', 1), ('sep_conv_3x3', 1), ('sep_conv_3x3', 2), ('sep_conv_3x3', 0), ('dil_conv_5x5', 3)], **normal_concat**=**range**(2, 6), **reduce**=[('max_pool_3x3', 0), ('dil_conv_3x3', 1), ('max_pool_3x3', 0), ('dil_conv_5x5', 2), ('avg_pool_3x3', 1), ('dil_conv_5x5', 3), ('avg_pool_3x3', 1), ('dil_conv_5x5', 4)], **reduce_concat**=**range**(2, 6)) |
| | #2 | **Genotype**(**normal**=[('dil_conv_3x3', 0), ('skip_connect', 1), ('dil_conv_5x5', 0), ('sep_conv_3x3', 1), ('max_pool_3x3', 0), ('sep_conv_3x3', 1), ('dil_conv_5x5', 3), ('dil_conv_5x5', 4)], **normal_concat**=**range**(2, 6), **reduce**=[('avg_pool_3x3', 0), ('max_pool_3x3', 0), ('dil_conv_5x5', 2), ('max_pool_3x3', 0), ('dil_conv_5x5', 3), ('dil_conv_5x5', 3), ('dil_conv_5x5', 4)], **reduce_concat**=**range**(2, 6)) |
| | #3 | **Genotype**(**normal**=[('dil_conv_3x3', 0), ('dil_conv_5x5', 1), ('dil_conv_5x5', 0), ('dil_conv_3x3', 2), ('skip_connect', 0), ('skip_connect', 1), ('sep_conv_3x3', 0), ('dil_conv_5x5', 3)], **normal_concat**=**range**(2, 6), **reduce**=[('avg_pool_3x3', 0), ('dil_conv_3x3', 1), ('avg_pool_3x3', 0), ('dil_conv_5x5', 2), ('max_pool_3x3', 1), ('skip_connect', 2), ('dil_conv_5x5', 2), ('dil_conv_5x5', 3)], **reduce_concat**=**range**(2, 6)) |
| S2 | #1 | **Genotype**(**normal**=[('sep_conv_3x3', 0), ('sep_conv_3x3', 1), ('sep_conv_3x3', 0), ('sep_conv_3x3', 1), ('sep_conv_3x3', 1), ('sep_conv_3x3', 2), ('sep_conv_3x3', 1), ('sep_conv_3x3', 4)], **normal_concat**=**range**(2, 6), **reduce**=[('skip_connect', 0), ('sep_conv_3x3', 1), ('sep_conv_3x3', 0), ('skip_connect', 2), ('sep_conv_3x3', 0), ('sep_conv_3x3', 2), ('skip_connect', 1), ('sep_conv_3x3', 3)], **reduce_concat**=**range**(2, 6)) |
| | #2 | **Genotype**(**normal**=[('sep_conv_3x3', 0), ('sep_conv_3x3', 1), ('sep_conv_3x3', 0), ('sep_conv_3x3', 2), ('sep_conv_3x3', 1), ('sep_conv_3x3', 3), ('skip_connect', 1), ('sep_conv_3x3', 4)], **normal_concat**=**range**(2, 6), **reduce**=[('sep_conv_3x3', 0), ('sep_conv_3x3', 1), ('sep_conv_3x3', 0), ('sep_conv_3x3', 2), ('skip_connect', 1), ('skip_connect', 3), ('sep_conv_3x3', 1), ('sep_conv_3x3', 3)], **reduce_concat**=**range**(2, 6)) |
| | #3 | **Genotype**(**normal**=[('sep_conv_3x3', 0), ('sep_conv_3x3', 1), ('sep_conv_3x3', 0), ('sep_conv_3x3', 1), ('skip_connect', 0), ('skip_connect', 1), ('skip_connect', 0), ('sep_conv_3x3', 3)], **normal_concat**=**range**(2, 6), **reduce**=[('skip_connect', 0), ('sep_conv_3x3', 1), ('sep_conv_3x3', 0), ('sep_conv_3x3', 1), ('sep_conv_3x3', 2), ('sep_conv_3x3', 3), ('skip_connect', 0), ('sep_conv_3x3', 1)], **reduce_concat**=**range**(2, 6)) |
| S3 | #1 | **Genotype**(**normal**=[('sep_conv_3x3', 0), ('sep_conv_3x3', 1), ('sep_conv_3x3', 0), ('skip_connect', 1), ('skip_connect', 0), ('sep_conv_3x3', 2), ('sep_conv_3x3', 0), ('sep_conv_3x3', 3)], **normal_concat**=**range**(2, 6), **reduce**=[('sep_conv_3x3', 0), ('sep_conv_3x3', 1), ('sep_conv_3x3', 1), ('sep_conv_3x3', 2), ('skip_connect', 0), ('sep_conv_3x3', 3), ('sep_conv_3x3', 0), ('sep_conv_3x3', 4)], **reduce_concat**=**range**(2, 6)) |
| | #2 | **Genotype**(**normal**=[('sep_conv_3x3', 0), ('sep_conv_3x3', 1), ('sep_conv_3x3', 0), ('sep_conv_3x3', 1), ('sep_conv_3x3', 0), ('sep_conv_3x3', 2), ('skip_connect', 0), ('skip_connect', 1)], **normal_concat**=**range**(2, 6), **reduce**=[('sep_conv_3x3', 0), ('sep_conv_3x3', 1), ('sep_conv_3x3', 0), ('skip_connect', 1), ('sep_conv_3x3', 1), ('sep_conv_3x3', 2), ('skip_connect', 0), ('skip_connect', 3)], **reduce_concat**=**range**(2, 6)) |
| | #3 | **Genotype**(**normal**=[('sep_conv_3x3', 0), ('sep_conv_3x3', 1), ('sep_conv_3x3', 0), ('skip_connect', 1), ('skip_connect', 0), ('sep_conv_3x3', 2), ('skip_connect', 0), ('sep_conv_3x3', 4)], **normal_concat**=**range**(2, 6), **reduce**=[('sep_conv_3x3', 0), ('sep_conv_3x3', 1), ('skip_connect', 0), ('sep_conv_3x3', 2), ('sep_conv_3x3', 0), ('sep_conv_3x3', 3), ('sep_conv_3x3', 2), ('skip_connect', 4)], **reduce_concat**=**range**(2, 6)) |
| S4 | #1 | **Genotype**(**normal**=[('sep_conv_3x3', 0), ('sep_conv_3x3', 1), ('sep_conv_3x3', 0), ('sep_conv_3x3', 2), ('noise', 0), ('sep_conv_3x3', 3), ('noise', 2), ('sep_conv_3x3', 4)], **normal_concat**=**range**(2, 6), **reduce**=[('sep_conv_3x3', 0), ('sep_conv_3x3', 1), ('sep_conv_3x3', 0), ('noise', 1), ('sep_conv_3x3', 1), ('noise', 3), ('noise', 0), ('sep_conv_3x3', 3)], **reduce_concat**=**range**(2, 6)) |
| | #2 | **Genotype**(**normal**=[('sep_conv_3x3', 0), ('sep_conv_3x3', 1), ('sep_conv_3x3', 0), ('sep_conv_3x3', 1), ('noise', 1), ('sep_conv_3x3', 3), ('sep_conv_3x3', 3), ('sep_conv_3x3', 4)], **normal_concat**=**range**(2, 6), **reduce**=[('sep_conv_3x3', 0), ('sep_conv_3x3', 1), ('sep_conv_3x3', 0), ('sep_conv_3x3', 2), ('sep_conv_3x3', 2), ('noise', 3), ('sep_conv_3x3', 0), ('noise', 1)], **reduce_concat**=**range**(2, 6)) |
| | #3 | **Genotype**(**normal**=[('sep_conv_3x3', 0), ('sep_conv_3x3', 1), ('sep_conv_3x3', 0), ('sep_conv_3x3', 2), ('noise', 1), ('sep_conv_3x3', 3), ('sep_conv_3x3', 0), ('sep_conv_3x3', 2)], **normal_concat**=**range**(2, 6), **reduce**=[('sep_conv_3x3', 0), ('sep_conv_3x3', 1), ('noise', 0), ('sep_conv_3x3', 2), ('sep_conv_3x3', 2), ('noise', 3), ('sep_conv_3x3', 1), ('noise', 2)], **reduce_concat**=**range**(2, 6)) |

Table 14: Architectures searched on DARTS image classification sub search space (SVHN).

| Setting | | Architecture |
|---|---|---|
| S1 | #1 | **Genotype**(**normal**=[('dil_conv_3x3', 0), ('dil_conv_5x5', 1), ('dil_conv_5x5', 0), ('dil_conv_3x3', 2), ('sep_conv_3x3', 1), ('sep_conv_3x3', 2), ('sep_conv_3x3', 0), ('dil_conv_3x3', 3)], **normal_concat**=**range**(2, 6), **reduce**=[('max_pool_3x3', 0), ('dil_conv_3x3', 1), ('avg_pool_3x3', 0), ('dil_conv_5x5', 2), ('max_pool_5x5', 3), ('dil_conv_5x5', 2), ('dil_conv_5x5', 3)], **reduce_concat**=**range**(2, 6)) |
| | #2 | **Genotype**(**normal**=[('dil_conv_3x3', 0), ('dil_conv_5x5', 1), ('dil_conv_5x5', 0), ('skip_connect', 1), ('sep_conv_3x3', 1), ('sep_conv_3x3', 2), ('dil_conv_3x3', 2), ('dil_conv_3x3', 3)], **normal_concat**=**range**(2, 6), **reduce**=[('max_pool_3x3', 0), ('max_pool_3x3', 1), ('max_pool_3x3', 0), ('dil_conv_5x5', 2), ('max_pool_3x3', 0), ('dil_conv_5x5', 3), ('avg_pool_3x3', 1), ('skip_connect', 4)], **reduce_concat**=**range**(2, 6)) |
| | #3 | **Genotype**(**normal**=[('dil_conv_3x3', 0), ('dil_conv_5x5', 1), ('dil_conv_5x5', 0), ('sep_conv_3x3', 1), ('skip_connect', 1), ('sep_conv_3x3', 2), ('skip_connect', 0), ('dil_conv_5x5', 3)], **normal_concat**=**range**(2, 6), **reduce**=[('max_pool_3x3', 0), ('max_pool_3x3', 1), ('max_pool_3x3', 0), ('dil_conv_5x5', 2), ('max_pool_3x3', 0), ('dil_conv_3x3', 2), ('dil_conv_5x5', 2), ('skip_connect', 3)], **reduce_concat**=**range**(2, 6)) |
| S2 | #1 | **Genotype**(**normal**=[('sep_conv_3x3', 0), ('skip_connect', 1), ('sep_conv_3x3', 0), ('sep_conv_3x3', 1), ('sep_conv_3x3', 0), ('sep_conv_3x3', 3), ('sep_conv_3x3', 0), ('skip_connect', 3)], **normal_concat**=**range**(2, 6), **reduce**=[('sep_conv_3x3', 0), ('sep_conv_3x3', 1), ('skip_connect', 0), ('sep_conv_3x3', 2), ('skip_connect', 2), ('skip_connect', 3), ('sep_conv_3x3', 2), ('sep_conv_3x3', 3)], **reduce_concat**=**range**(2, 6)) |
| | #2 | **Genotype**(**normal**=[('skip_connect', 0), ('skip_connect', 1), ('sep_conv_3x3', 0), ('sep_conv_3x3', 2), ('skip_connect', 0), ('sep_conv_3x3', 1), ('sep_conv_3x3', 3), ('sep_conv_3x3', 4)], **normal_concat**=**range**(2, 6), **reduce**=[('sep_conv_3x3', 0), ('sep_conv_3x3', 1), ('sep_conv_3x3', 0), ('sep_conv_3x3', 2), ('sep_conv_3x3', 2), ('sep_conv_3x3', 3), ('sep_conv_3x3', 0), ('sep_conv_3x3', 3)], **reduce_concat**=**range**(2, 6)) |
| | #3 | **Genotype**(**normal**=[('sep_conv_3x3', 0), ('skip_connect', 1), ('sep_conv_3x3', 0), ('sep_conv_3x3', 2), ('skip_connect', 0), ('sep_conv_3x3', 3), ('sep_conv_3x3', 3), ('skip_connect', 4)], **normal_concat**=**range**(2, 6), **reduce**=[('sep_conv_3x3', 0), ('sep_conv_3x3', 1), ('sep_conv_3x3', 1), ('sep_conv_3x3', 2), ('sep_conv_3x3', 1), ('skip_connect', 1), ('sep_conv_3x3', 3), ('skip_connect', 4)], **reduce_concat**=**range**(2, 6)) |
| S3 | #1 | **Genotype**(**normal**=[('sep_conv_3x3', 0), ('skip_connect', 1), ('sep_conv_3x3', 1), ('sep_conv_3x3', 2), ('skip_connect', 0), ('skip_connect', 2), ('sep_conv_3x3', 0), ('skip_connect', 1)], **normal_concat**=**range**(2, 6), **reduce**=[('sep_conv_3x3', 0), ('sep_conv_3x3', 1), ('sep_conv_3x3', 0), ('sep_conv_3x3', 2), ('sep_conv_3x3', 0), ('sep_conv_3x3', 2), ('sep_conv_3x3', 1), ('sep_conv_3x3', 3)], **reduce_concat**=**range**(2, 6)) |
| | #2 | **Genotype**(**normal**=[('sep_conv_3x3', 0), ('skip_connect', 1), ('sep_conv_3x3', 0), ('sep_conv_3x3', 2), ('sep_conv_3x3', 0), ('sep_conv_3x3', 1), ('skip_connect', 0), ('sep_conv_3x3', 3)], **normal_concat**=**range**(2, 6), **reduce**=[('skip_connect', 0), ('sep_conv_3x3', 1), ('skip_connect', 1), ('sep_conv_3x3', 2), ('sep_conv_3x3', 0), ('sep_conv_3x3', 3), ('sep_conv_3x3', 2), ('sep_conv_3x3', 4)], **reduce_concat**=**range**(2, 6)) |
| | #3 | **Genotype**(**normal**=[('sep_conv_3x3', 0), ('skip_connect', 1), ('sep_conv_3x3', 0), ('sep_conv_3x3', 2), ('sep_conv_3x3', 3), ('sep_conv_3x3', 2), ('skip_connect', 3)], **normal_concat**=**range**(2, 6), **reduce**=[('sep_conv_3x3', 0), ('sep_conv_3x3', 1), ('skip_connect', 1), ('skip_connect', 2), ('sep_conv_3x3', 1), ('skip_connect', 3), ('skip_connect', 1), ('sep_conv_3x3', 2)], **reduce_concat**=**range**(2, 6)) |
| S4 | #1 | **Genotype**(**normal**=[('sep_conv_3x3', 0), ('sep_conv_3x3', 1), ('sep_conv_3x3', 0), ('noise', 2), ('sep_conv_3x3', 0), ('noise', 3), ('sep_conv_3x3', 1), ('sep_conv_3x3', 3)], **normal_concat**=**range**(2, 6), **reduce**=[('sep_conv_3x3', 0), ('noise', 1), ('sep_conv_3x3', 1), ('noise', 2), ('noise', 0), ('sep_conv_3x3', 2), ('sep_conv_3x3', 2), ('sep_conv_3x3', 4)], **reduce_concat**=**range**(2, 6)) |
| | #2 | **Genotype**(**normal**=[('sep_conv_3x3', 0), ('sep_conv_3x3', 1), ('sep_conv_3x3', 0), ('noise', 1), ('sep_conv_3x3', 0), ('sep_conv_3x3', 3), ('noise', 2), ('sep_conv_3x3', 3)], **normal_concat**=**range**(2, 6), **reduce**=[('sep_conv_3x3', 0), ('sep_conv_3x3', 1), ('noise', 1), ('sep_conv_3x3', 2), ('sep_conv_3x3', 2), ('sep_conv_3x3', 3), ('sep_conv_3x3', 4)], **reduce_concat**=**range**(2, 6)) |
| | #3 | **Genotype**(**normal**=[('sep_conv_3x3', 0), ('noise', 1), ('sep_conv_3x3', 0), ('sep_conv_3x3', 2), ('noise', 0), ('noise', 3), ('sep_conv_3x3', 2), ('sep_conv_3x3', 4)], **normal_concat**=**range**(2, 6), **reduce**=[('sep_conv_3x3', 0), ('sep_conv_3x3', 1), ('sep_conv_3x3', 0), ('sep_conv_3x3', 2), ('sep_conv_3x3', 0), ('sep_conv_3x3', 2), ('sep_conv_3x3', 3)], **reduce_concat**=**range**(2, 6)) |

Table 15: Architectures searched on DARTS PTB search space.

| Setting | Architecture |
|---|---|
| Run1 | `Genotype(recurrent=[('tanh', 0), ('relu', 1), ('sigmoid', 1), ('relu', 1), ('tanh', 0), ('sigmoid', 0), ('tanh', 4), ('identity', 0)], concat=range(1, 9))` |
| Run2 | `Genotype(recurrent=[('relu', 0), ('tanh', 1), ('tanh', 1), ('relu', 3), ('relu', 1), ('tanh', 5), ('relu', 5), ('tanh', 4)], concat=range(1, 9))` |
| Run3 | `Genotype(recurrent=[('tanh', 0), ('relu', 0), ('tanh', 1), ('relu', 1), ('sigmoid', 4), ('tanh', 2), ('tanh', 0), ('tanh', 0)], concat=range(1, 9))` |
| Run4 | `Genotype(recurrent=[('tanh', 0), ('relu', 1), ('identity', 1), ('relu', 2), ('sigmoid', 2), ('identity', 4), ('relu', 1), ('relu', 0)], concat=range(1, 9))` |

Table 16: NASBench validation accuracy across the 36 settings for EPS. "#i" stands for the $i_{th}$ search run.

| Setting | #1 | #2 | #3 | #4 | #5 | Avg. Acc. |
|---|---|---|---|---|---|---|
| $P$:64, $S$:32, $M$:1, $I\_val$:50 | 91.21 | 88.90 | 91.21 | 91.30 | 90.85 | $90.69 \pm 0.91$ |
| $P$:64, $S$:32, $M$:1, $I\_val$:100 | 90.84 | 91.53 | 91.14 | 90.24 | 91.22 | $90.99 \pm 0.44$ |
| $P$:64, $S$:32, $M$:1, $I\_val$:200 | 91.00 | 89.15 | 90.54 | 88.23 | 91.51 | $90.09 \pm 1.21$ |
| $P$:64, $S$:32, $M$:4, $I\_val$:50 | 91.05 | 91.21 | 91.21 | 91.50 | 90.93 | $91.18 \pm 0.19$ |
| $P$:64, $S$:32, $M$:4, $I\_val$:100 | 91.19 | 90.78 | 90.85 | 89.90 | 90.93 | $90.73 \pm 0.44$ |
| $P$:64, $S$:32, $M$:4, $I\_val$:200 | 91.05 | 91.00 | 91.18 | 91.19 | 91.42 | $91.17 \pm 0.15$ |
| $P$:64, $S$:64, $M$:1, $I\_val$:50 | 91.02 | 90.85 | 90.92 | 89.84 | 91.03 | $90.73 \pm 0.45$ |
| $P$:64, $S$:64, $M$:1, $I\_val$:100 | 91.03 | 90.81 | 91.48 | 88.90 | 90.84 | $90.61 \pm 0.89$ |
| $P$:64, $S$:64, $M$:1, $I\_val$:200 | 90.99 | 91.12 | 91.22 | 90.87 | 91.02 | $91.04 \pm 0.12$ |
| $P$:64, $S$:64, $M$:4, $I\_val$:50 | 90.93 | 91.35 | 91.55 | 91.55 | 91.48 | $91.37 \pm 0.23$ |
| $P$:64, $S$:64, $M$:4, $I\_val$:100 | 91.55 | 90.92 | 91.21 | 91.55 | 90.84 | $91.22 \pm 0.30$ |
| $P$:64, $S$:64, $M$:4, $I\_val$:200 | 90.55 | 90.99 | 91.35 | 90.93 | 89.31 | $90.63 \pm 0.71$ |
| $P$:128, $S$:32, $M$:1, $I\_val$:50 | 91.01 | 91.36 | 90.42 | 91.42 | 89.37 | $90.72 \pm 0.76$ |
| $P$:128, $S$:32, $M$:1, $I\_val$:100 | 91.45 | 91.06 | 90.59 | 91.37 | 91.28 | $91.15 \pm 0.31$ |
| $P$:128, $S$:32, $M$:1, $I\_val$:200 | 91.12 | 84.69 | 91.51 | 91.02 | 91.19 | $89.91 \pm 2.61$ |
| $P$:128, $S$:32, $M$:4, $I\_val$:50 | 91.30 | 91.42 | 89.59 | 91.03 | 91.28 | $90.93 \pm 0.68$ |
| $P$:128, $S$:32, $M$:4, $I\_val$:100 | 91.55 | 90.93 | 91.61 | 91.34 | 91.36 | $91.36 \pm 0.24$ |
| $P$:128, $S$:32, $M$:4, $I\_val$:200 | 91.19 | 91.35 | 90.55 | 91.42 | 91.01 | $91.11 \pm 0.31$ |
| $P$:128, $S$:64, $M$:1, $I\_val$:50 | 91.03 | 90.93 | 91.19 | 91.53 | 88.77 | $90.69 \pm 0.98$ |
| $P$:128, $S$:64, $M$:1, $I\_val$:100 | 91.30 | 90.79 | 91.46 | 91.02 | 91.36 | $91.19 \pm 0.25$ |
| $P$:128, $S$:64, $M$:1, $I\_val$:200 | 88.88 | 84.23 | 91.55 | 90.79 | 88.62 | $88.81 \pm 2.55$ |
| $P$:128, $S$:64, $M$:4, $I\_val$:50 | 89.54 | 91.42 | 91.50 | 91.19 | 89.08 | $90.55 \pm 1.03$ |
| $P$:128, $S$:64, $M$:4, $I\_val$:100 | 91.30 | 91.21 | 91.30 | 90.77 | 91.03 | $91.12 \pm 0.20$ |
| $P$:128, $S$:64, $M$:4, $I\_val$:200 | 91.16 | 91.55 | 90.87 | 90.95 | 90.87 | $91.08 \pm 0.26$ |
| $P$:256, $S$:32, $M$:1, $I\_val$:50 | 91.03 | 90.84 | 88.94 | 91.42 | 91.40 | $90.73 \pm 0.92$ |
| $P$:256, $S$:32, $M$:1, $I\_val$:100 | 88.87 | 90.16 | 91.12 | 91.30 | 90.34 | $90.36 \pm 0.86$ |
| $P$:256, $S$:32, $M$:1, $I\_val$:200 | 90.51 | 90.90 | 91.07 | 84.15 | 91.19 | $89.56 \pm 2.72$ |
| $P$:256, $S$:32, $M$:4, $I\_val$:50 | 91.42 | 91.22 | 91.28 | 91.18 | 90.84 | $91.19 \pm 0.19$ |
| $P$:256, $S$:32, $M$:4, $I\_val$:100 | 91.00 | 91.12 | 91.00 | 90.84 | 90.87 | $90.97 \pm 0.10$ |
| $P$:256, $S$:32, $M$:4, $I\_val$:200 | 90.77 | 91.03 | 91.22 | 91.01 | 91.21 | $91.05 \pm 0.16$ |
| $P$:256, $S$:64, $M$:1, $I\_val$:50 | 91.50 | 91.30 | 91.53 | 91.55 | 88.90 | $90.96 \pm 1.03$ |
| $P$:256, $S$:64, $M$:1, $I\_val$:100 | 91.44 | 91.04 | 91.03 | 91.00 | 91.21 | $91.15 \pm 0.16$ |
| $P$:256, $S$:64, $M$:1, $I\_val$:200 | 91.48 | 90.77 | 91.30 | 90.84 | 90.90 | $91.06 \pm 0.28$ |
| $P$:256, $S$:64, $M$:4, $I\_val$:50 | 91.55 | 91.42 | 91.50 | 91.42 | 91.61 | $91.50 \pm 0.07$ |
| $P$:256, $S$:64, $M$:4, $I\_val$:100 | 91.28 | 91.42 | 91.42 | 89.23 | 91.28 | $90.93 \pm 0.85$ |
| $P$:256, $S$:64, $M$:4, $I\_val$:200 | 91.03 | 91.36 | 90.99 | 90.93 | 90.93 | $91.05 \pm 0.16$ |

Table 17: DARTS search space testing accuracy. "#i" stands for the $i_{th}$ search run. "run i" stands for the $i_{th}$ validation training run.

| Settings | | #1 | #2 | #3 | #4 | #5 |
|---|---|---|---|---|---|---|
| | run 1 | 97.59 | 97.23 | 97.30 | 97.54 | 97.52 |
| CIFAR-10 | run 2 | 97.40 | 97.38 | 97.19 | 97.38 | 97.30 |
| | run 3 | 97.46 | 97.38 | 97.14 | 97.26 | 97.32 |
| | run 1 | 81.71 | 82.28 | 81.33 | 82.60 | 82.54 |
| CIFAR-100 | run 2 | 80.99 | 81.49 | 81.51 | 82.22 | 83.10 |
| | run 3 | 81.66 | 82.28 | 82.15 | 82.57 | 83.07 |

Table 18: DARTS sub search space testing accuracy. "#i" stands for the $i_{th}$ search run. "run i" stands for the $i_{th}$ validation training run.

| Setting | | CIFAR-10 | | | CIFAR-100 | | | SVHN | | |
|---|---|---|---|---|---|---|---|---|---|---|
| | | #1 | #2 | #3 | #1 | #2 | #3 | #1 | #2 | #3 |
| S1 | run 1 | 97.04 | 97.08 | 97.44 | 78.26 | 75.59 | 75.11 | 97.61 | 97.52 | 97.42 |
| | run 2 | 97.01 | 97.05 | 97.35 | 77.46 | 75.69 | 75.24 | 97.50 | 97.56 | 97.51 |
| | run 3 | 97.02 | 97.13 | 97.36 | 77.92 | 75.82 | 74.19 | 97.58 | 97.45 | 97.43 |
| S2 | run 1 | 96.83 | 96.84 | 96.98 | 78.35 | 77.96 | 77.83 | 97.35 | 97.51 | 97.52 |
| | run 2 | 96.74 | 96.70 | 96.81 | 77.76 | 78.06 | 77.70 | 97.21 | 97.33 | 97.50 |
| | run 3 | 96.68 | 96.73 | 96.84 | 79.14 | 78.18 | 77.74 | 97.43 | 97.56 | 97.55 |
| S3 | run 1 | 97.53 | 97.37 | 97.32 | 78.42 | 77.97 | 77.13 | 97.38 | 97.58 | 97.56 |
| | run 2 | 97.41 | 97.26 | 97.32 | 78.18 | 76.94 | 77.27 | 97.43 | 97.43 | 97.53 |
| | run 3 | 97.44 | 97.37 | 97.50 | 78.62 | 77.70 | 76.38 | 97.38 | 97.51 | 97.46 |
| S4 | run 1 | 96.55 | 95.48 | 96.39 | 75.61 | 76.65 | 76.18 | 97.18 | 97.40 | 97.18 |
| | run 2 | 96.14 | 96.21 | 96.40 | 75.19 | 76.90 | 76.88 | 97.18 | 97.25 | 97.07 |
| | run 3 | 96.49 | 95.79 | 96.18 | 74.60 | 76.43 | 76.19 | 97.17 | 97.30 | 97.27 |

