# OpenReview forum: "Improving Random-Sampling Neural Architecture Search by Evolving the Proxy Search Space"
_ICLR.cc/2021/Conference — Reject_

### Official Review · AnonReviewer2 · 2020-10-25
**Official Blind Review #2**

**Rating:** 6
**Confidence:** 2

**Review:**

Summary:\
This paper finds an assumption in neural architecture search (NAS) where "performance estimated in a weight-sharing setting translates to actualy achievable performance" only holds for uniformly sampled architecture but breaks down when sampling from top-performing architectures. As a result a new sampling method based on tournament selection for NAS is proposed.
-----
+Strengths\
+Novel insights are revealed for NAS work which may be useful for future research.
+Clear experiments are devised to demonstrate the insights.
+Strong experiment and ablation results compared to prior NAS approaches.
-----
-Concerns\
-My biggest question here would be how the best architecture from EPS compare to other NAS-based models that are successfully adopted in a wide range of vision application, such as EFficientNet. It would be interesting to show performance on larger dataset.
-I am also interested to know the range of performance between best and worst for top 100/60/20% of models to better understand how poorly correlated top architectures in GS actually translate to loss in performance beyond performance of the final architecture.
-----
Recommendation\
To my best knowledge, this paper uncovers some interesting insight and show performance improvements. My main concern is the viability of the proposed structure in more practical vision problems such as recognition on larger dataset. My recommendation is leaning towards weak accept.

---

> ### Author Response · Authors · 2020-11-16
> **Reply to Reviewer#2**
>
> Thank you very much for your very detailed and very useful review. We summarize your questions and reply to you:
> Q1. Experiment on large dataset
> We admit that we have not run our models on a larger dataset like ImageNet yet. The concentration of our work is to study the drawbacks of RandomNAS and use the EPS to overcome them. Hence, we choose a direction to study several computationally lite cases that are friendly for repetition and reproduction. Multiple repeats can be done for each experiment, and we can see the theoretic performance gain by different NAS algorithms rather than the engineering gains on ImageNet. In our work, we explored NASBench-201, DARTS search space (CIFAR-10 and CIFAR-100),  4 DARTS sub search space (CIFAR-10, CIFAR-100, and SVHN), and DARTS NLP search space with EPS. The total 16 cases help us further study the RandomNAS and support the EPS can find good architectures in various computer vision tasks and even the NLP task. Since our work is reproducible and easy to follow, we open-sourced our code for the community for future study.
> Q2. We summarize the statistic of NASBench-201 (CIFAR-10) in the table below:
>
> |                    |Validation  acc.	                  |Test acc.                                 |
>
> |-----------------|---------|---------|---------|---------|---------|---------|---------|---------|
>
> |                    |Lowest|Highest|Median|Mean||Lowest|Highest|Median|Mean|
>
> |-----------------|---------|---------|---------|---------|---------|---------|---------|---------|
>
> |Top 100%    |9.71     |91.61  |87.27   |83.65  |10.00  |94.37  |90.71  |87.04  |
>
> |Top 60%      |86.31   |91.61  |88.71   |88.58  |89.52   |94.37  |92.02  |91.85  |
>
> |Top 20%      |89.13   |91.61  |89.64   |89.80  |92.36   |94.37  |92.93  |92.98  |
>
> Also, we plotted the histogram in the link below for better understanding:
> https://drive.google.com/file/d/16m9USGB58-gS4TA2FjGGHPUEdjw3beTG/view?usp=sharing

---

### Official Review · AnonReviewer3 · 2020-10-28
**Please explain the novelty and motivation clearly**

**Rating:** 4
**Confidence:** 3

**Review:**

Motivated by exploring the ranking correlations of the existing RandomNAS in NASBench-201, this paper proposes EPS to improve the search efficiency and keep good ranking correlations by evolving the proxy search space (PS) in RandomNAS. Specially, EPS contains three stages: 1) training the supernet in PS, 2) validating the architectures among the PS and 3) evolving the PS by tournament selection with the aging mechanism. Furthermore, a model-size-based regularization is introduced in the selection stage. Experiments on some popular benchmarks demonstrate the effectiveness of the method.

Strengths
1) The paper is well written and easy to follow. The algorithm procedure is clearly provided and the code is released.
2) Some empirical evidences are provided to explain the limitations of the existing RandomNAS.

Weaknesses
1) Lack of novelty. One the one hand, EPS seems a combination of [1] with a weight-sharing supernet, and the differences 2,3,4 with CARS do not convince me well. Except for the motivation and the fitness in EA, EPS and CARS are highly similar. One the other hand, the intuition of the solutions for the limitations which the authors presented in Sec. 2 is not clearly provided. From my view, Q_pop in Algorithm 1 is same with the population in EA methods. The proxy search space is not clearly explained.
2) How to choose the 4000 architectures from RandomNAS in Fig. 2(a)?
3) EPS validates each architecture only on a single batch. Dose one batch validation bring biases to the performance and the ranking evaluation of the architectures? Does the phenomenons in Sec. 2 are caused by one batch validation due to the biases? Comparison with Full batches validation (the whole validation set) should be considered.
4) Results on PTB are not promising. RandomNAS (RSPS) achieves better perplexity with less search cost.
5) “32 x 5 runs” and “32 settings” should be “36 x 5 runs” and “36 settings”, respectively.

The novelty and the similarity with previous works are my main concerns. I am currently leaning towards a negative score but would like to see the authors' responses and other reviewer's comments.

[1] Esteban Real, Alok Aggarwal, Yanping Huang, and Quoc V Le. Regularized evolution for image classifier architecture search. In AAAI, 2019.

---

> ### Author Response · Authors · 2020-11-16
> **Reply to Reviwer#3**
>
> Thank you very much for your very detailed and very useful review. We summarize your questions and hope our answers can address your concerns.
>
> Q1. The incremental novelty.
> The difference between CARS and EPS is as you mentioned we have a very solid motivation from the observations of RandomNAS. Also, we pointed out that the aging mechanism is critical for the EPS in Table 1, while CARS doesn’t use the aging mechanism. On the other hand, our proposed size regularization is a simple yet effective way for the improvement of RandomNAS ranking correlation. Besides the proposed EPS, our observation of the RandomNAS drawbacks equally contribute to the paper’s novelty. To the best of our knowledge, we are the first to do the detailed study for RandomNAS and both the observations and the new methods can help the AutoML community for further exploration.
>
> Q2. The intuition of the solutions
> We’d like to appreciate your concerns. Here we use simple words to re-describe the two observations we present in Section 2:
> 1. The original RandomNAS trained on the global search space is hard to get a correct ranking for a set of “good” architectures.
> 2. The original RandomNAS tends to take architectures with smaller sizes as the better ones.
> We hope the above descriptions can help you better understand the overall Section 2. And we welcome the comments if you have new advice.
>
> Q3. Q_pop
> Q_pop is the population of the EPS. Utilizing the aging mechanism, Q_pop is considered as a queue with architectures first in first out.
>
> Q4.  The proxy search space
> proxy search space(PS) is a subset with a certain number of architectures sampled from the global search space(GS)
>
>
> Q5. How to choose the 4000 architectures from RandomNAS?
> The 4000 architectures are randomly sampled from the global search space. The overall 25.6% portion is representative for the global search space.
>
> Q6. Batch validation vs. full validation
> First, all the experiments in Section 2 use full validation. To address the concern of the gap between batch validation and full validation, we calculate the best ranking correlation achieved using the batch validation using RandomNAS. The full validation ranking correlation is 0.78 and the batch validation correlation is 0.74. Since the full validation on 102400 architectures (80000(iteration)/50(interval)*64(validated architectures)) requires 80+ GPU hrs. For the EPS search algorithm, we use the batch validation for a trade-off between the correlation and the speed.
>
>
> Q8. Results on PTB
> 1. In another work (Zhang et al. 2020) they reduplicated the RandomNAS and only achieves 59.7 valid perplexity.
> 2. Also, we followed the RandomNAS work to do the 4 runs comparison of 300 epochs training and summarize the results as follow:
>
> |Method|run1|run2|run3|run4|Avg.|
> |----------|------|------|------|------|-----------|
> |DARTS|67.3| 66.3| 63.4| 63.4|65.1|
> |RSPS|66.3| 64.6 | 64.1 | 63.8 |  64.7|
> |EPS|65.10|63.74*|64.72|64.77| 64.58|
>
> It shows that the average perplexity of architectures found by EPS surpass both DARTS and RSPS. We added it to the appendix in the revised version.
> Q9. Some typos
> Thank you for pointing out, we fixed it in the revised version.

---

### Official Review · AnonReviewer1 · 2020-10-28
**Review of Improving Random-Sampling Neural Architecture Search**

**Rating:** 5
**Confidence:** 4

**Review:**

This paper claims that random search-based NAS methods show a low ranking correlation among top-20% candidate architectures in the search phase. To address this issue, this paper proposes to introduce a proxy search space consisting of good architectures and evolve it using evolutionary algorithms. This paper also proposes a simple size regularization to help the NAS algorithm escape from the small architecture traps. The experimental results show that the proposed approach achieves competitive performance with baseline methods.

Pros
- This paper analyzes the behavior in the random search-based NAS in detail and proposes a new strategy based on the observation to tackle the issue.

Cons
- The proposed method should be compared with recent NAS methods to clarify the contribution of this paper. Many related studies are missing.
- The details of the algorithm of EPS are unclear. For example, it's not clear what exactly is being done in the RandomInitArch and Mutate operations. Also, what is the reason for using sample_set instead of population queue?
- What do you mean by the population is a proxy search space on page 4? Does the proxy search space mean a specific architecture without over parameterization or an over parameterized architecture? Please elaborate on the definition of the proxy search space and each population.
- In section 2, the authors give an analysis on random search-based NAS, but does this hold for other conditions such as for the search space used in DARTS and Robust DARTS?

Overall, the analysis in this paper is interesting, but there are some unclear points to be clarified for publication as mentioned above.

---

> ### Author Response · Authors · 2020-11-16
> **Reply to Reviewer#1**
>
> We thank the reviewer for the very detailed and useful comments. Below we provide responses to each concern.
> Q1: May missing study with recent NAS methods?
> We sincerely thank you again for your concern. Also, we’d like to humbly point out that, in the work, we did explore NASBench-201, DARTS search space (CIFAR-10 and CIFAR-100),  4 DARTS sub search space (CIFAR-10, CIFAR-100, and SVHN), and DARTS NLP search space with EPS. We choose RandomNAS as the baseline and compare our algorithm with several recent NAS on the above 16 cases. EPS achieves state-of-the-art results on most of them. The results are shown in Table 2-6.
>
> Q2: The details of the algorithm of EPS are unclear. For example, it's not clear what exactly is being done in the RandomInitArch and Mutate operations. Also, what is the reason for using sample_set instead of population queue?
> 1. RandomInitArch is to randomly, uniformly sample architecture from the global search space.
> Thank you for pointing out the unclear definition, we made a definition of the item in Section 3 in the revised version.
> 2. Mutate operations:
> For NASBench-201, the rule for mutation is: The current operation on the edge has a fixed probability to be mutated to a new operation(including itself) on the same edge.
> For the DARTS search space family, the rules for mutation are 1. The current operation on the edge has a fixed probability to be mutated to a new operation(including itself) on the same edge. 2. If a node has unconnected predecessors, one of its edges has a fixed probability to switch to an edge linked to the unconnected predecessors, and the operation on the new edge will be chosen randomly.
> We have the definition of the mutate operation in Appendix B.
> Thank you again for pointing out the vague definition, we put the reference in Section 3 in the revised version and our open-sourced code can also help better understand and reduplicate the work.
> 3. The notion of the sample set comes from the tournament selection. Using a small sample set can improve validation efficiency. Also, we run different settings with sample set size in {32,64} and population size in {64,128,256} on NASBench-201 for the detailed study. The results are shown in Table 14.
>
> Q3: The definitions of population and the proxy search space.
> proxy search space(PS) is a subset with a certain number of architectures sampled from the global search space(GS). We have the definition of PS in the introduction and we re-define it cleanly in Section 2 in the revised version for better understanding. Thank you again for pointing it out.
> The population is initialized with P architectures uniformly sampled from the global search space. So it’s a proxy search space.
> Q4: In section 2, the authors give an analysis of random search-based NAS, but does this hold for other conditions such as for the search space used in DARTS and Robust DARTS?
> Since DARTS search space contains 10^18 architectures which is hard for the full training for each architecture from the scratch, we provide several pieces of evidence support that the assumption holds for the DARTS and Robust DARTS:
> 1.  In the paper, we did an experiment that randomly sampled 27 architectures from the last validation interval (after 80,000 training iterations) for EPS in the DARTS search space. Then each architecture is trained three times from scratch and the mean of the three accuracies are considered as the ground truth. We show the Spearman’s ρ between the ranking of ground truth and EPS’ prediction, the ρ between the ranking of ground truth and RandomNAS’s prediction in Figure 4. We observed that the architectures sampled from the final proxy search space surpass the random sampling baseline by a large margin (the average accuracy of samples 95.54% vs. 95.15%).  Moreover, EPS delivers 0.68 Spearman’s ρ while RandomNAS performs worse (0.41) at distinguishing the difference between them. The observation supports the assumption that the final proxy search space consists of good architectures and EPS can deliver a higher correlation compared with the RandomNAS in GS.
> 2. The extensive experimental results show that EPS can find better architectures than RandomNAS on both DARTS and DARTS sub search space (please see Table 3-6).

---

### Official Review · AnonReviewer4 · 2020-10-28
**The paper aims to bring the help of search strategies to construct the approximate search space.**

**Rating:** 5
**Confidence:** 4

**Review:**

This paper proposes Evolving the Proxy Search Space (EPS) as a new RandomNAS-based approach. The goal is to find an effective proxy search space (PS) that is only a small subset of GS to dramatically improve RandomNAS’s search efficiency while at the same time keeping a good correlation for the top-performing architectures. EPS runs in three stages iteratively: Training the supernet by randomly sampling from a PS; Validating the architectures among the PS on a subset of the validation dataset in the training interval; Evolving the PS by a tournament selection evolutionary algorithm with the aging mechanism.


The paper is well written and easy to follow. The idea of efficiently sampling from the search space sounds interesting and the paper aims to bring the help of search strategies to construct the search space by itself. However, it is kind of incremental work since the EPS (Algorithm 1) is exactly similar to the aging evolution in Real et al. (2018, 2019) and this paper is using this search strategy to gradually build the search space on the fly.

I think any of the existing search strategies 1) random search 2) Evolutionary algorithms (Real et al. (2018)) 4) progresive decision process (PNAS; Liu, et al 2018), etc can be used to find a proxy search space. While a random sample from GS is simply random search, proposed EPS is exactly the evolutionary strategy to build search space. One may even use progressive NAS algorithm as a proxy search space!  Comparing how these different proxy search spaces improve efficiency will improve the novelty and make the paper more strong.

---

> ### Author Response · Authors · 2020-11-16
> **Reply to Reviewer#4**
>
> Thank you very much for your very detailed and useful review. We summarize your questions and reply to you:
> Q1: The similarity between the EPS and the aging evolution in Real et al.
> The motivation for using the aging mechanism is different from the Real et al. Here's the reason: 1) in a weight-sharing supernet, a young architecture may have a higher loss and be removed before well trained.  2) an old architecture survives in the population when it performs well in the early stage and produces many mutations, which may dominate the population and mislead the search direction. We compared EPS with the one w/o aging in Table 1, which indicates that the aging mechanism is necessary for the EPS. Another difference is Real et al. (2019) trained every visited architecture from scratch while EPS utilizes the weight-sharing supernet and leads to a search time similar to one architecture's training time. Hence, Real et al. take more than 9,000X consumption than ours.
> Besides the proposed EPS, we are the first to look into the details of the RandomNAS and our observation of the RandomNAS drawbacks equally contribute to the paper’s novelty. Also, our proposed size regularization is a simple yet effective way for the improvement of RandomNAS ranking correlation.
> Q2: Can other NAS algorithms utilize the proxy search space?
> To address the question that if other NAS algorithms (e.g., NAS in a progressive way) can help build a better proxy search space than the EPS, we choose two alternative algorithms in the NASBench-201 for a detailed comparison:
> 1. Progressive search: Similar to the work Progressive DARTS[1], we start with a supernet with N = 1. (N is defined in Figure 1 in the NAS-BENCH-201 paper, the number of searchable cells are 3N.) And we gradually increase N by 1 every 15000 iterations as the interval until N = 5 (default setting in the benchmark). Other settings are adopted from EPS.
> 2. MLP for the exploration: Instead of using the criterion of mutating the k architectures with the lowest loss, we introduce an MLP to learn the loss of architectures in the population and predict other architectures' performance.
> The table below shows the five-run experiment results in NASBench-201. Original EPS outperforms the other two methods on CIFAR-10. Since Progressive EPS speeds up ~1.5x, we considered it as a good trade-off between the performance and the speed.
>
> |                   |Val acc.|Test acc.|
> |----------------|----------|------------|
> |EPS            |91.50   |94.34     |
> |Progressive|91.33   |94.00     |
> |MLP            |89.17  |92.57     |
>
> We updated the details of the above 2 algorithms’  implementation in the Appendix and uploaded the source code to the Github repo.
> [1] Chen, Xin, et al. "Progressive differentiable architecture search: Bridging the depth gap between search and evaluation." Proceedings of the IEEE International Conference on Computer Vision. 2019.

---

### Decision · Program_Chairs · 2021-01-07
**Final Decision**

**Decision:**

Reject

**Comment:**

The paper analyzes the behavior of random search-based NAS and provided new insights (e.g., a low ranking correlation among top-20% candidate architectures in the search phase). An extensive set of experiments were also conducted. However, most reviewers found the incremental nature and similarity with previous works to be a concern. I would encourage the authors to better position their work and better explain the novel methodological aspects.